



# Evaluation of Wetland CH$_4$ in the JULES Land Surface Model Using Satellite Observations

Robert J. Parker[1,2], Chris Wilson[3,4], Edward Comyn-Platt[5,6], Garry Hayman[6], Toby R. Marthews[6], A. Anthony Bloom[7], Mark F. Lunt[8], Nicola Gedney[9], Simon J. Dadson[6,10], Joe McNorton[5], Neil Humpage[1,2], Hartmut Boesch[1,2], Martyn P. Chipperfield[3,4], Paul I. Palmer[8,11], and Dai Yamazaki[12]

[1]National Centre for Earth Observation, University of Leicester, UK
[2]Earth Observation Science, School of Physics and Astronomy, University of Leicester, UK
[3]National Centre for Earth Observation, University of Leeds, UK
[4]School of Earth and Environment, University of Leeds, UK
[5]European Centre For Medium-Range Weather Forecasts, Reading, UK
[6]UK Centre for Ecology & Hydrology, Wallingford, UK
[7]Jet Propulsion Laboratory, California Institute of Technology, Pasadena, CA, USA
[8]School of GeoSciences, University of Edinburgh, Edinburgh, UK
[9]Met Office Hadley Centre, Joint Centre for Hydrometeorological Research, Maclean Building, Wallingford, UK
[10]School of Geography and the Environment, University of Oxford, Oxford, UK
[11]National Centre for Earth Observation, University of Edinburgh, Edinburgh, UK
[12]Global Hydrological Forecast Center, Institute of Industrial Science, The University of Tokyo, Tokyo, Japan

**Correspondence:** R. J. Parker (rjp23@le.ac.uk)

**Abstract.**

Wetlands are the largest natural source of methane. The ability to model the emissions of methane from natural wetlands accurately is critical to our understanding of the global methane budget and how it may change under future climate scenarios. The simulation of wetland methane emissions involves a complicated system of meteorological drivers coupled to hydrological

5    and biogeochemical processes. The Joint UK Land Environment Simulator (JULES) is a process-based land surface model that underpins the UK Earth System Model and is capable of generating estimates of wetland methane emissions.

In this study we use GOSAT satellite observations of atmospheric methane along with the TOMCAT global 3-D chemistry transport model to evaluate the performance of JULES in reproducing the seasonal cycle of methane over a wide range of tropical wetlands. By using an ensemble of JULES simulations with differing input data and process configurations, we investigate

10    the relative importance of the meteorological driving data, the vegetation, the temperature dependency of wetland methane production and the wetland extent. We find that JULES typically performs well in replicating the observed methane seasonal cycle. We calculate correlation coefficients to the observed seasonal cycle of between 0.58 to 0.88 for most regions, however the seasonal cycle amplitude is typically underestimated (by between 1.8 ppb and 19.5 ppb). This level of performance is comparable to that typically provided by state-of-the-art data-driven wetland CH$_4$ emission inventories. The meteorological

15    driving data is found to be the most significant factor in determining the ensemble performance, with temperature dependency and vegetation having moderate effects. We find that neither wetland extent configuration out-performs the other but this does lead to poor performance in some regions.



We focus in detail on three African wetland regions (Sudd, Southern Africa and Congo) where we find the performance of JULES to be poor and explore the reasons for this in detail. We find that neither wetland extent configuration used is sufficient in representing the wetland distribution in these regions (underestimating the wetland seasonal cycle amplitude by 11.1 ppb, 19.5 ppb and 10.1 ppb respectively, with correlation coefficients of 0.23, 0.01 and 0.31). We employ the CaMa-Flood model to explicitly represent river and floodplain water dynamics and find these JULES-CaMa-Flood simulations are capable of providing wetland extent more consistent with observations in this regions, highlighting this as an important area for future model development.

## 1  Introduction

Methane ($CH_4$) is a significant greenhouse gas, with a global warming potential (GWP) many times greater than that of $CO_2$ (Etminan et al. (2016), 100-year GWP = 28). According to IPCC et al. (2021, in press), methane accounts for approximately 20% of the increase in radiative forcing from pre-industrial to present-day. The relatively short atmospheric lifetime of methane ($\sim$9 years, Prather et al. (2012)) means that reductions provide significant potential for mitigation of climate change to help address the goals of the Paris Agreement (O'Connor et al., 2010; Ganesan et al., 2019). However, the global methane budget is highly complex with a range of natural and anthropogenic sources (Saunois et al., 2020), many of which are still poorly-constrained and possess large uncertainties (Dlugokencky et al., 2009; Nisbet et al., 2014).

Wetlands are the largest natural methane source and are comparable (or larger) in magnitude than emissions from agriculture/waste and fossil fuels (Saunois et al., 2020). Natural wetlands are inundated ecosystems with water-saturated soil or peat and include permanent or seasonal floodplains, swamps, marshes and peatlands where the anaerobic conditions lead to $CH_4$ production via methanogenic bacteria. Importantly, the uncertainty in $CH_4$ emissions from wetlands remains one of the most significant challenges for understanding the global $CH_4$ budget. Not only are there large uncertainties on processes and mechanisms related to the $CH_4$ emission itself (Melton et al., 2013), but the wetland extent is highly uncertain (Bloom et al., 2010; Kirschke et al., 2013; Stocker et al., 2014) as is the response to meteorological drivers (Poulter et al., 2017; Parker et al., 2018)

One important step in better understanding the global $CH_4$ budget is reconciling the bottom-up estimates of $CH_4$ emissions (e.g. from land surface models) with top-down estimates based on atmospheric observations. The latest assessment of the global $CH_4$ budget (Saunois et al., 2020) has a bottom-up estimate of wetland $CH_4$ emissions of 149 Tg $CH_4$ $yr^{-1}$ (range 102 - 182) compared to a top-down estimate of 181 Tg $CH_4$ $yr^{-1}$ (range 159 - 200) for 2008-2017. Recent work (Folberth et al., 2022, submitted) has coupled wetland $CH_4$ emissions from the Joint UK Land Environment Simulator (JULES) into the UK Earth System Model (UKESM) for the first time, allowing interactive wetland emissions from JULES to be used in climate simulations. To fully exploit this new capability, it is vital that the performance of the JULES wetland $CH_4$ scheme is well-characterised and evaluated against present-day observations.

In this study we perform an evaluation of the wetland $CH_4$ emissions from the JULES land surface model using satellite observations of atmospheric $CH_4$ columns in order to both assess the utility of the model in providing emission estimates



as well as to diagnose any discrepancies against observations that may lead to future model improvements and increased understanding of the relevant processes.

The objectives of this study are:

- To provide an evaluation of the performance of JULES wetland $CH_4$ simulations across the tropics using satellite remote sensing data.

- To evaluate and characterise the differences in performance across an ensemble of JULES simulations with different configurations and identify the best-performing configuration(s) with the most suitable input data.

- To explain the underlying reasons for poorly-performing regions, relating these to the processes within JULES and provide guidance on potential improvements.

In Section 2 we introduce the JULES land surface model, explain how wetland methane emissions are calculated and describe
the ensemble of simulations that we have produced. Section 3 details the datasets and tools used to directly compare the JULES $CH_4$ emissions to observations. In Section 4 we perform an evaluation of the seasonal cycle of JULES $CH_4$ emissions over a range of wetland regions and in Section 5 we focus in more detail on the challenging African regions. We conclude the study in Section 6.

## 2  JULES Wetland $CH_4$ Emissions

The Joint UK Land Environment Simulator, JULES (Best et al., 2011; Clark et al., 2011), is a process-based land surface model that both underpins the UK Earth System Model (Sellar et al., 2019) and acts as a standalone model capable of simulating many processes related to the land surface by describing the carbon, water and energy exchanges. We use JULES version 5.1 in this study.

### 2.1  Generation of Wetlands within JULES

TOPMODEL (TOPography-based hydrological MODEL) is a rainfall-runoff model where estimates of surface and subsurface runoff are produced taking into account the topography of the land surface (Beven, 2012). This is defined through the topographic index, which is related to the relative propensity for soil saturation in that it incorporates both slope and upstream area. TOPMODEL was originally applied at the scale of small catchments, using pixels smaller than 50 m x 50 m in extent, but this framework has since been extended to global applications at a much wider range of spatial scales (Marthews et al., 2015;
Gedney et al., 2019). TOPMODEL remains one of the most popular and widely-used runoff production models (Beven et al., 2021) and has been implemented within the framework of the JULES model for many years (Best et al., 2011).

TOPMODEL is implemented in JULES as part of the large-scale hydrology scheme (Gedney and Cox, 2003; Best et al., 2011). A deep layer of restrictive water flow, added to the bottom of the standard soil column at a 3m depth, results in the production of a saturated soil zone and a water table. The water table moves vertically when the soil moisture changes. Within





each grid box the statistical distribution of topographic index (Marthews et al., 2015) is combined with the mean water table depth. This enables the simulation of a sub-grid water table distribution and therefore the extent of wetland in the grid box.

## 2.2 JULES Wetland CH₄ Emissions

The JULES land surface model calculates methane wetland emissions $F_{\mathrm{CH_4}}$, from three key factors, namely the amount of available substrate carbon, the temperature and the inundated area below the water table (Gedney et al., 2004; Clark et al.,

85 2011):

$$F_{\mathrm{CH_4}} = k_{\mathrm{CH_4}} \cdot f_w \sum_{i=1}^{n\,C_s\,pools} \kappa_i \cdot \sum_{z=0m}^{z=3m} e^{-\gamma z} \cdot C_{s_{i,z}} \cdot Q_{10}(T_{soil})^{0.1(T_{soil}-T_0)} \tag{1}$$

$k_{CH4}$ is a dimensionless scaling constant ($7.41 \times 10^{-12}$) for wetland CH₄ emissions when soil carbon is taken as the substrate for CH₄ emissions. The wetland fraction (i.e. the proportion of a grid cell where the water table is at/above the surface, and below a threshold indicative of significant flow (Gedney et al., 2004)) is denoted by $f_w$. $z$ is the depth of soil column (in m),

$i$ is the soil carbon pool, $\kappa_i$ (s⁻¹) is the specific respiration rate of each pool (Table 8 of Clark et al. (2011)), $C_s$ (kg $m^{-2}$) is soil carbon and $T_{soil}$ (K) is the soil temperature, averaged over the soil layers in the top 1 m of soil. The decay constant $\gamma$ (= 0.4 m⁻¹) describes the reduced contribution of CH₄ emission at deeper soil layers due to inhibited transport and increased oxidation through overlaying soil layers. This representation of inhibition is a simplification. However, previous work which explicitly represented these processes showed little to no improvement when compared with in-situ observations (McNorton

et al., 2016). We do not model CH₄ emissions from freshwater lakes.

## 2.3 JULES Ensemble Experimental Setup

As outlined in Equation 1, there are a variety of options within JULES and choices of input data which affect the calculation of CH₄ from wetlands, which we call a 'configuration'. In this study, we produce an ensemble of JULES simulations that span a range of configurations. Different configurations allow adjustment of factors that have all been identified as key sources

of uncertainty in previous wetland methane modelling efforts. We identify which is/are the optimal configuration(s) through comparison of model outputs against observations. The JULES ensemble that we produce comprises: 2 different sets of meteorological driving data (ERA-Interim and WFDEI, Sect. 2.3.1), 3 different vegetation configurations (prescribed phenology, dynamic vegetation with and without competition, Sect. 2.3.2), 2 different temperature dependencies ($Q_{10} = 3.7$ and $Q_{10} = 5.0$, Sect. 2.3.3) and 2 different wetland extent parameterisations (the default from JULES and a version masked via the Sur-

face WAter Microwave Product Series (SWAMPS) wetland extent, Sect. 2.3.4). This results in an ensemble with 24 members (2x3x2x2). In order to identify ensemble members, we assign to each member a 4-digit ID as shown in Figure 1. Thus the ensemble member using WFDEI meteorology data (2), using dynamic vegetation (3), with the lower temperature dependency (1) and with the original JULES wetland extent (1) is ensemble member 2311.





In a post-processing step, the time series of annual wetland emissions of each ensemble member is separately scaled to give annual emissions of 180 Tg $CH_4$ $yr^{-1}$ for the year 2000 (Saunois et al., 2016), as described in Comyn-Platt et al. (2018).

Maps of the $CH_4$ emissions for each ensemble member are presented in Figure 2 for August 2011. Clear differences are observed relating to the different ensemble configurations, including: substantial differences between ERA-Interim and WFDEI-based ensemble members with the magnitude of the emissions in the WFDEI members visibly smaller; and large spatial differences based on the Default vs SWAMPS wetland extent masking, with SWAMPS significantly reducing the wetland areas and concentrating the emissions, particularly removing the widespread but low emissions found more generally in the Default members.

**4-digit code describes ensemble member - ABCD**

| A | 1 | 2 | |
|---|---|---|---|
| Met Driving Data | ERA-Interim | WFDEI | |

| B | 1 | 2 | 3 |
|---|---|---|---|
| Vegetation | Phenology, 9pfts | TRIFFID Fixed, 9pfts | TRIFFID Dynamic, 9pfts |

| C | 1 | 2 | |
|---|---|---|---|
| Temperature Dependence | q10 = 3.7 | q10 = 5.0 | |

| D | 1 | 2 | |
|---|---|---|---|
| Extent Parameterisation | JULES | JULES with SWAMPS mask | |

**Figure 1.** Description of the 24 JULES ensemble members used in this study, comprising of 2 x Meteorological Driving Data, 3 x Vegetation configurations, 2 x Temperature Dependencies and 2 x Wetland Extent configurations. The 4-digit code (ABCD) is used to identify the individual ensemble members.

### 2.3.1 Driving Data: ERA-Interim vs WFDEI

Meteorological forcing data is used to drive the JULES land surface model. The meteorological parameters used in this study are: air temperature, surface pressure, precipitation, short and long-wave radiation, relative humidity and wind speed. In the ensemble we use two sources for the meteorological data, ERA-Interim and WFDEI.

The ERA-Interim Reanalysis (Dee et al., 2011) is a widely used global atmospheric reanalysis product produced by the European Centre for Medium-Range Weather Forecasts (ECMWF). The WATCH Forcing Data ERA-Interim (WFDEI) is based on the ERA-Interim Reanalysis data but includes the modifications as outlined in (Weedon et al., 2014). Namely, interpolation to a 0.5° x 0.5° resolution, a sequential elevation correction and a monthly bias correction based on observations.

**Figure 2.** Example (August 2011) of wetland CH$_4$ emissions generated from each JULES ensemble members used in this study. The ensemble comprises of 2 x Meteorological Driving Data configurations, 2 x Wetland Extent configurations, 2 x Temperature Dependency configurations and 3 x Vegetation configurations. Each panel is labelled with the details of its configuration, following the format of the key (bottom-left).

### 2.3.2 Vegetation

Vegetation is represented by nine plant functional types (PFTs): broadleaf deciduous trees, tropical broadleaf evergreen trees, temperate broadleaf evergreen trees, needle-leaf deciduous trees, needle-leaf evergreen trees, C3 and C4 grasses, deciduous and evergreen shrubs (Harper et al., 2016). Depending on the options chosen, these PFTs can be in competition for space, based on the TRIFFID (Top-down Representation of Interactive Foliage and Flora Including Dynamics) dynamic vegetation module within JULES (Clark et al., 2011). There are also four non-vegetated surface types: urban, water, bare soil and ice.





The ensemble uses three different JULES configurations to describe the vegetation behaviour (Clark et al., 2011): a configuration based on calculating leaf-level phenology and two configurations based on the TRIFFID dynamic vegetation module in JULES, with and without vegetation competition (i.e. allowing for changes in surface coverage by different plant functional types or not, respectively). The calculation of leaf phenology is independent of the calculation of the evolution of vegetation coverage and is available even when the TRIFFID dynamic vegetation module is not used.

The number of carbon pools used in Equation 1 depends on the soil biogeochemistry model (soil_bgc_model) and vegetation options selected. For the leaf phenology vegetation option, soil_bgc_model = 1 and a single carbon pool is used. For the vegetation configurations using the TRIFFID dynamic vegetation model, soil_bgc_model = 2 and four carbon pools are used based on the Roth-C model (Clark et al., 2011).

### 2.3.3 Temperature Dependence: $Q_{10}$ = 3.7 vs 5.0

As indicated in Equation 1, the $CH_4$ emission is strongly dependent on the temperature of the soil. This temperature dependency of methanogenesis is generally parameterised using a $Q_{10}$ value that approximates the Arrhenius equation. As discussed in Gedney et al. (2004), the approach that JULES takes due to applying this approximation globally over a wide temperature range is to use an effective or generalised $Q_{10}$ that fits the form of the Arrhenius equation exactly (Equation 2).

$$Q_{10}(T) = Q_{10}(T_0)^{T_0/T} \tag{2}$$

### 2.3.4 Wetland Extent: JULES vs JULES with SWAMPS mask

JULES generates wetland extent following the TOPMODEL approach as outlined in Section 2.1. As accurate wetland extent is one of the largest challenges in relation to modelling wetland emissions of methane (Saunois et al., 2020), the ensemble also provides an alternative observationally-constrained wetland extent. In this instance, the JULES wetland area is simply masked by the SWAMPS dataset (Schroeder et al., 2015), meaning that any wetland extent that is inconsistent with the SWAMPS observations is disregarded.

## 3   Datasets Used for Comparing JULES $CH_4$ Emissions to Atmospheric Observations

### 3.1   GOSAT $CH_4$ Observations

The primary observational dataset that we use for evaluation of the JULES $CH_4$ is the University of Leicester GOSAT Proxy $XCH_4$ (Parker et al., 2011, 2020a). The GOSAT satellite, launched in 2009 by the Japanese Space Agency, was the first dedicated greenhouse gas observing satellite (Kuze et al., 2009). This data was recently used (Parker et al., 2020b) to evaluate the WetCHARTs $CH_4$ emission database (Bloom et al., 2017a) and has previously been used for many wetland-related studies including Parker et al. (2015); Berchet et al. (2015); McNorton et al. (2016); Lunt et al. (2019); Saunois et al. (2020); Wilson et al. (2021); Maasakkers et al. (2021); Lunt et al. (2021).





GOSAT measures the signal of reflected sunlight in the shortwave infrared (SWIR) and as such is capable of providing measurements over land and also over the ocean in cases where sun-glint reflection allows. The GOSAT Proxy XCH$_4$ retrieval provides around 15k-25k observations over land each month and, after changes to the sun-glint sampling in 2015, a comparable number over the ocean. For a full description of the data, including evaluation and validation, see Parker et al. (2020a).

### 3.2    TOMCAT Atmospheric CH$_4$ Simulations

In order to link surface CH$_4$ emissions as generated by JULES with atmospheric observations as measured by GOSAT, it is necessary to run the emissions through a global chemistry transport model.

In this study, we use the TOMCAT 3-D model (Chipperfield, 2006), ran globally between 2009 and 2017 at 1.125° horizontal resolution and 60 vertical levels up to 0.1 hPa. The model setup is consistent with that in Parker et al. (2020b). In short, non-wetland CH$_4$ fluxes are taken from the EDGAR v4.2 database for anthropogenic emissions and the GFED v4.1s dataset for

biomass burning emissions. Annually repeating rice paddy emissions are used from Yan et al. (2009), with ocean and termite sources used following Patra et al. (2011). The atmospheric (OH, O($^1$D) and stratospheric Cl) and soil sinks are as described in McNorton et al. (2016).

For the wetland CH$_4$ fluxes, the emissions generated for each of the 24 JULES ensemble members (Section 2.3) are assigned to individual tracers. These tracers each contain the wetland and non-wetland CH$_4$ fluxes and therefore an additional tracer

containing no wetland emissions is used as a reference to remove the non-wetland effects.

## 4    Evaluation of JULES Wetland CH$_4$ Seasonal Cycle

In this section we evaluate the seasonal cycle of the wetland CH$_4$ emissions generated from the ensemble of JULES simulations against atmospheric satellite observations. We perform the same analysis on the JULES wetland emission datasets as was used for the evaluation of the WetCHARTs emission dataset Parker et al. (2020b), thereby enabling comparison of results and

conclusions.

The evaluation is performed over 7 large-scale areas (Global, Northern Hemisphere, Southern Hemisphere, 60°S-60°N, Tropics, North Tropics, South Tropics) as well as 16 specific wetland areas as indicated in Figure 3.

To calculate the XCH$_4$ seasonal cycle, we apply the NOAA CurveFitting routine (Thoning et al., 1989; NOAA) to the GOSAT CH$_4$ observations as well as the TOMCAT model simulations for each of the JULES wetland emission ensemble

members. To determine the wetland-specific signal, we apply the same technique to the TOMCAT tracer that contained no wetland emissions and subtract that signal. This results (Figure 4) in a wetland XCH$_4$ seasonal cycle for each region from GOSAT and from each of the model ensemble members. The observed (GOSAT) seasonal cycle magnitude varies significantly between regions (e.g. contrast the Pantanal to East Amazon) and can also be seen to vary strongly between years for the same region (e.g. contrast S.E. Asia for 2010 to 2017). Qualitatively, the ensemble of JULES-based simulations are not dissimilar to

the observations, however the simulated seasonal cycles are typically weaker in magnitude than the observations. Although the ensemble spread can be large in some regions (e.g. Indo-Gangetic), the regions with a strong observed seasonal cycle typically





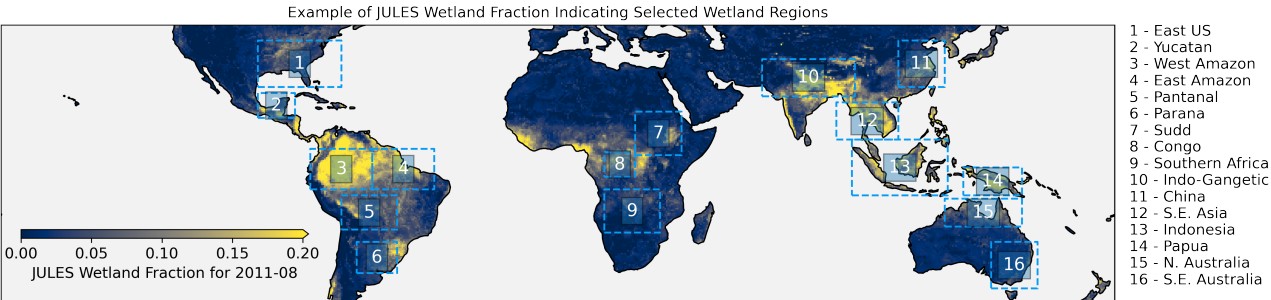

**Figure 3.** Map showing the locations of the 16 wetland regions considered in this study. A representative month (August 2011) of the JULES wetland fraction is shown as the basemap.

exhibit a strong seasonal cycle in the JULES ensemble, albeit typically with a smaller magnitude. This suggests overall, JULES is generally capable of reproducing the region-to-region and month-to-month wetland emissions that we see from observations but details for specific regions can fail to match the observations.





**Figure 4.** Time series showing the GOSAT (red) and JULES ensemble (blue, min/max envelope) wetland $CH_4$ seasonal cycles for 7 large-scale areas and 16 specific wetland regions. The wetland seasonal cycle is calculated by subtracting the TOMCAT model simulations that do not contain any wetland emissions. For each time series, the dashed horizontal lines indicate the [-25, 0, 25] as indicated in the bottom panel.

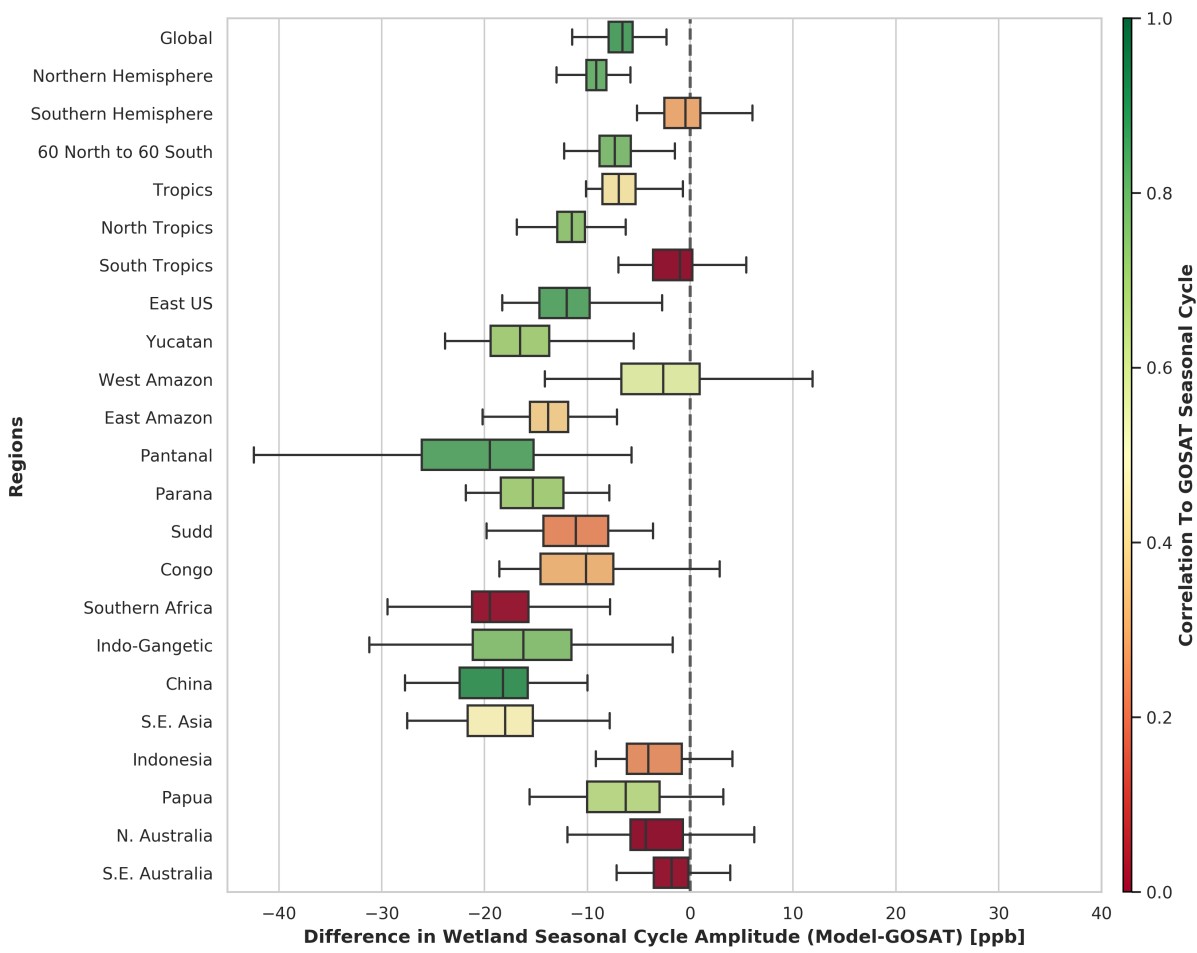

**Figure 5.** Boxplot showing the distribution of the difference in the wetland CH$_4$ seasonal-cycle amplitude between the JULES ensemble and GOSAT observations for all years (2009 - 2017). A box-and-whisker (box: quartiles, whiskers: min/max) is calculated for each of the regions (7 large-scale areas and 16 specific wetland areas) and is coloured by the mean value of the correlation coefficient between the modelled and observed wetland CH$_4$ seasonal cycle.





A more rigorous quantitative evaluation of the seasonal cycle phase and magnitude is shown in Figure 5. In this analysis we produce a box-and-whisker plot for the distribution of the model-GOSAT wetland $XCH_4$ seasonal cycle amplitude differences ($\Delta A$), combining all ensemble members and all years for each region. Further, the box is coloured according to the mean value of the correlation coefficient ($R_{cycle}$) between the GOSAT and model seasonal cycles.

Globally we find that the JULES ensembles underestimate the $XCH_4$ wetland seasonal cycle amplitude by approximately 6.6 ppb (quartiles: 5.6 ppb - 7.9 ppb) with a correlation coefficient of 0.85. When considering the northern and southern hemispheres we see somewhat different behaviour, with $\Delta A$ of -9.2 ppb and -0.4 ppb respectively. This north-south difference is exaggerated further when contrasting the North Tropics ($\Delta A$ = -11.5 ppb, $R_{cycle}$ = 0.73) and the South Tropics ($\Delta A$ = -1.0 ppb, $R_{cycle}$ = 0.0).

When focusing on specific wetland regions we find that the evaluation is varied and performance is very region-dependent. For example, although $R_{cycle}$ = 0.83 for the Pantanal region suggesting that the phase of the seasonal cycle is reasonably well-captured, the seasonal cycle amplitude is significantly underestimated ($\Delta A$ = -19.5 ppb) and furthermore, this underestimation has a very large spread between ensemble members and years (ranging from -42.4 ppb to -5.7 ppb). In contrast, the Paraná region has a slightly poorer $R_{cycle}$ (0.70) and slightly better $\Delta A$ (-15.3 ppb) but with significantly smaller spread between ensemble members (-21.8 ppb to -7.9 ppb).

For the majority of wetland regions (East US, Yucatan, West Amazon, Pantanal, Paraná, Indo-Gangetic, China, Papua) $R_{cycle}$ shows a reasonable correlation of between 0.58 to 0.88. However, several regions stand out as having a particularly poor $R_{cycle}$ value (East Amazon, Sudd, Congo, Southern Africa, Indonesia, N. Australia, S.E. Australia). This poor correlation coefficient is easily explained for some regions (especially the Australian regions) where the seasonal cycle itself is very small (Figure 4). However, of particular note are the three African regions (Sudd, Congo and Southern Africa) where the seasonal cycle itself can be relatively strong but timing is in poor agreement between the JULES ensemble and the observations ($R_{cycle}$ values: 0.23, 0.31, 0.01 respectively). We revisit these regions in Section 5 and perform a more detailed evaluation in order to explain the poor performance here.

Despite these few poorly-performing regions, JULES shows reasonable-to-good performance overall in representing the observed seasonal cycle. It is informative here to judge the performance of JULES against the current state-of-the-art wetland emission dataset, WetCHARTs. In Parker et al. (2020b) we evaluated the performance of WetCHARTs in the same way as we evaluate JULES here so a direct comparison of the ability to model the observed seasonal cycle can be made. We reproduce Figure 4 from Parker et al. (2020b) in the appendix to this work (Figure A1) and contrast it against Figure 5. Overall the comparisons for the different wetland regions are largely in agreement, with a strikingly similar distribution in $\Delta A$ between regions. Both WetCHARTs and JULES typically underestimate the wetland seasonal cycle magnitude with the largest $\Delta A$ occuring in the same regions (Southern Africa, Indo-Gangetic, China, S.E. Asia). The largest discrepancies between the JULES analysis and our previous WetCHARTs analysis are that: for WetCHARTs the ensemble spread ($\sigma_A$) in the Congo is far larger than for JULES, while $R_{cycle}$ is reasonable compared to poor for JULES; although the biases for Southern African are very similar, $R_{cycle}$ for WetCHARTs is reasonable while again, it is poor for JULES. The above all suggests that the performance





of JULES is very comparable to that of the observation-driven WetCHARTs emissions, albeit with some differences in key
regions.

## 4.1   Attribution of Performance to Specific Configuration Choices

A significant feature apparent in the analysis so far is that the spread in $\Delta A$ across the ensemble members is typically large,
often in excess of 20 ppb between the minimum/maximum $\Delta A$ values. Understanding which ensemble members perform
well (and poorly) is an important step towards identifying which parameters and processes are driving the discrepancies to
observations. To investigate this, we calculate the *change* in two metrics, the correlation coefficient between the GOSAT and
modelled wetland seasonal cycle ($R_{cycle}$) and the standard deviation of the seasonal cycle amplitude($\sigma_A$), *above* the minimum
value for that metric. We denote these changes as $\Delta R_{cycle}$ and $\Delta \sigma_A$. We do this for the different ensemble parameter groupings
(meteorological driving data, vegetation, temperature dependency, wetland extent) individually and hold the other parameters
constant. To elaborate, out of the 24 ensemble members, the ensemble is split into (2x3x2x2) groupings (see Section 2.3 and
Figure 1). Using the meteorological driving data as an example, there are 12 different configurations that use ERA-Interim
and 12 configurations that use WFDEI. We compare the statistics for the performance of these configurations for pairs of
configurations where the only difference is which meteorological driving data is used and calculate the *change* in the metric
between the highest and lowest values. We then do likewise for the other parameters (vegetation, temperature dependency
and wetland extent). Note that for vegetation there are 3 configuration possibilities (phenology, fixed-TRIFFID and dynamic-
TRIFFID) and this results in triplets rather than pairs of members that are compared.

The results of this analysis are presented in Figure 6 with all regions collated into a single set of results. The ensemble
members driven by WFDEI consistently out-perform the ERA-Interim based members with both a significantly higher $\Delta R_{cycle}$
(a median increase of 0.12 with quartile values of 0.02 and 0.24) and significantly lower $\Delta \sigma_A$ (a median decrease of 0.53 ppb).
For the vegetation configurations, the results are more mixed without any single configuration being substantially better than
the rest but the phenology-based configurations do exhibit a slightly higher $\Delta R_{cycle}$ (0.03, 0.01 and 0.01 for Phenology,
TRIFFID-Fixed and TRIFFID-Dynamic) and lower $\Delta \sigma_A$ than the TRIFFID configurations suggesting that overall it performs
slightly better. However the significant overlapping spread here suggests that these results are much more region-dependent.
For temperature dependency the lower $Q_{10}$ value (3.7) performs better than the higher $Q_{10}$ value (5.0) but again, the spread
in both $\Delta R_{cycle}$ (e.g. 75th-percentile values of 0.15 and 0.09 for a $Q_{10}$ of 3.7 and 5.0 respectively) and $\Delta \sigma_A$ (75th-percentile
values of 0.53 ppb and 0.72 ppb) are high, suggesting a large region-to-region variability (consistent with Turetsky et al.
(2014) who measured a wide range of $Q_{10}$ values across different wetland types). Finally the choice of wetland extent between
JULES and SWAMPS is found to make little difference with SWAMPS very slightly increasing the correlation and decreasing
the standard deviation over the original JULES. We discuss this aspect in more detail below.

Overall we can conclude that the source of the meteorological driving data (ERA-Interim vs WFDEI) is the most significant
factor in how well JULES is able to reproduce the wetland seasonal cycle with WFDEI performing (almost) unanimously
better than ERA-Interim over the 16 wetland regions that we consider. This highlights the importance of the meteorological
data and the value in the interpolation and bias-correction that is performed as part of the WFDEI methodology (Weedon





et al., 2014). The choice of the vegetation and temperature dependency configurations were found to improve (or worsen) the representation of the seasonal cycle depending on their choice but this was found to be much more region-dependent with

a greater spread. Perhaps surprisingly, the choice of wetland extent configuration was found to have less of an effect when collating results across all regions. However, an important point to make here is that we are solely comparing the performance between two extent configurations and find that neither is significantly better than the other. This does not preclude extent itself from being important. It should also be remembered here that for the majority of regions, $R_{cycle}$ already shows a good correlation to observations for the majority of ensemble members (see Figure 5) implying that the extent is already sufficiently

well-reproduced in these regions. In the following section we focus on case studies over the 3 poorly-performing African wetland regions and demonstrate the significance of poorly reproducing wetland extent in these regions.

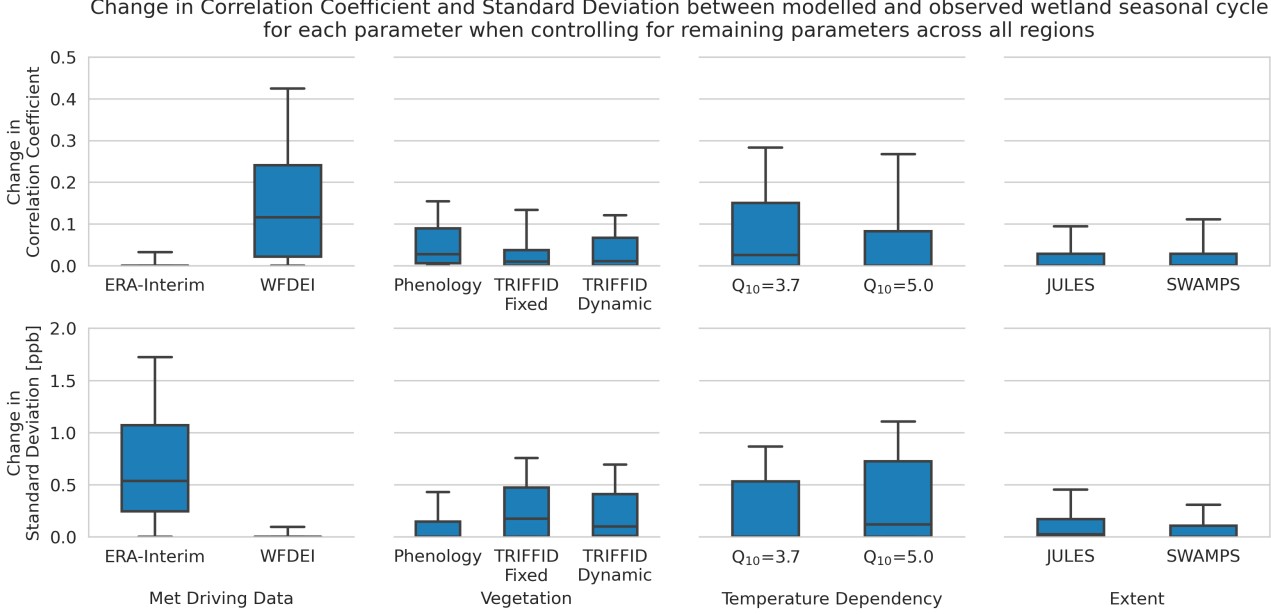

**Figure 6.** The *change* in correlation coefficient (top panel) and standard deviation (bottom panel) between the JULES ensemble and GOSAT wetland CH$_4$ seasonal cycle when controlling for the remaining ensemble parameters. The change is the difference above the minimum value for each *set* of ensemble members. An increased correlation coefficient should be considered an improvement, whereas an increased standard deviation should be considered a worsening. The changes are calculated for all 16 wetland regions in this study and presented as a box-and-whisker plot (box: quartiles, whiskers: min/max).





## 5  Evaluation of JULES Ensemble Over Africa Wetland Regions

We now investigate three significant African wetland regions (the Sudd, the Congo and Southern Africa) in detail and evaluate the performance of the JULES wetland methane emission estimates in these regions.

Figure 7 presents the same analysis as performed in Figure 6 but broken down individually for the three African wetland regions. Overall, the same general pattern that we find for all regions persists individually for these regions but with some interesting exceptions.

For the meteorological data, the WFDEI ensemble members show improved $\Delta R_{cycle}$ (0.26, 0.12 and 0.46 medians for Sudd, Congo and Southern Africa respectively) with ERA-Interim worsening the $\Delta \sigma_A$ value (by 0.60 ppb, 0.50 ppb and 1.35 ppb

respectively). As a reminder here, a value of 0 (as is the case for the change in ERA-Interim), indicates that the selection consistently performs the same (be that the lowest correlation coefficient or the smallest standard deviation) in relation to the other possible selection(s).

For the vegetation configuration, as found across all regions combined, there is not a distinctly better configuration. The phenology-based ensemble members perform best for the Sudd, with the highest $\Delta R_{cycle}$ (0.12) and lowest $\Delta \sigma_A$ (0.0 ppb,

indicating that it consistently out-performs the other configurations). However, for the Congo region there seems to be very little improvement, or indeed variability, between the 3 different vegetation options. This is largely expected due to low variability/seasonality in the tropical broadleaf vegetation. For the Southern Africa region, the dynamic TRIFFID configuration performs slightly worse than the others ($\Delta \sigma_A$ increasing by 0.28 ppb) but the performance of phenology and fixed-TRIFFID is hard to differentiate.

The temperature dependency exhibits very strong regional behaviour. For example, for Southern Africa the temperature dependency can improve $\Delta R_{cycle}$ by 0.75 for the lower $Q_{10}$ value versus the higher value and at the same time, the higher $Q_{10}$ value can worsen the $\Delta \sigma_A$ by over 1.6 ppb. In contrast, for the Congo the higher $Q_{10}$ value improves $\Delta R_{cycle}$ by 0.32 with the lower $Q_{10}$ value worsening $\Delta \sigma_A$ by over 0.6 ppb. While this does not leave us with a clear indication that one $Q_{10}$ value is universally better than the other, it does highlight the potential for significantly improving the $\Delta R_{cycle}$ by selection of

appropriate region-specific values. It should be noted that while some studies (e.g. Turetsky et al. (2014)) have measured a wide variability in $Q_{10}$ values across different wetland types (e.g. bog, fen, swamps), these have typically focused on subtropical, temperature and northern high-latitude regions. Further observations and constraints on the temperature dependency of tropical wetlands would be useful in this context.

Finally, neither configuration of wetland extent is found to significantly out-perform the other for any of the 3 regions. For

the Congo and Southern Africa regions, there is very little difference and very little improvement from selecting one extent configuration over the other. For the Sudd region there is a slightly larger spread across the ensembles but that is true for both the JULES and SWAMPS wetland extent configurations with neither significantly improving $\Delta R_{cycle}$ or worsening the $\Delta \sigma_A$.





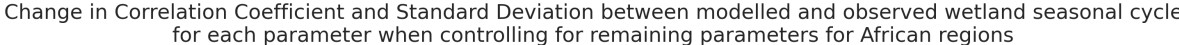

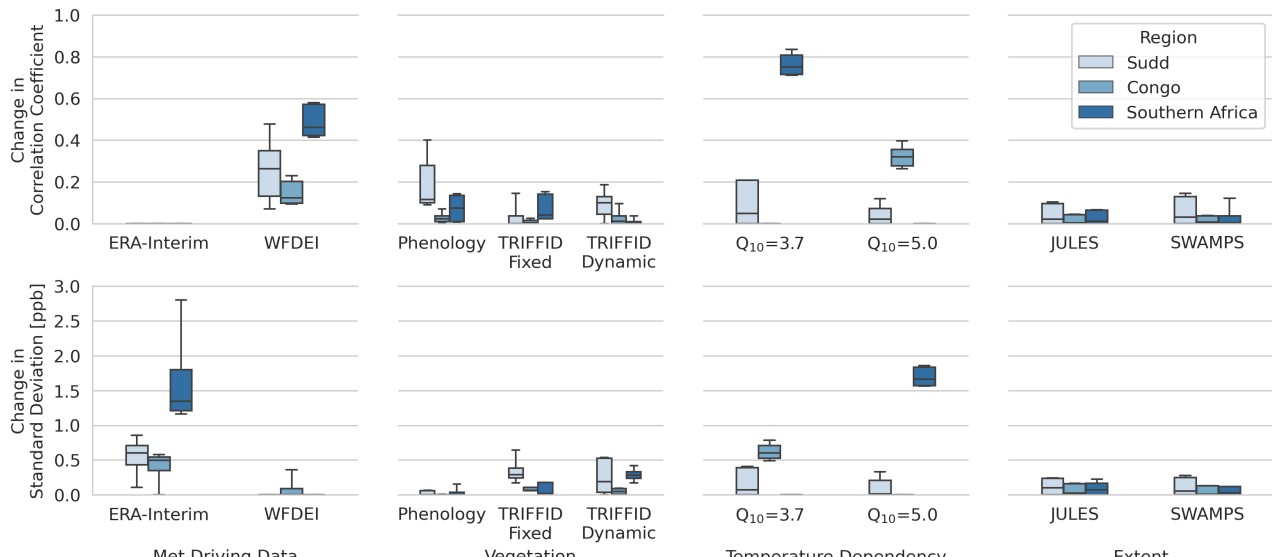

**Figure 7.** The *change* in correlation coefficient (top panel) and standard deviation (bottom panel) between the JULES ensemble and GOSAT wetland CH$_4$ seasonal cycle when controlling for the remaining ensemble parameters. The change is the difference above the minimum value for each set of ensemble members. An increased correlation coefficient should be considered an improvement, whereas an increased standard deviation should be considered a worsening. The figure shows the changes for the 3 wetland regions we examine over Africa (Sudd, Congo and Southern Africa) and are presented as box-and-whisker plots (box: quartiles, whiskers: min/max).



## 5.1 Additional Datasets For African Case Study Analysis

We find that several additional datasets offer utility in further diagnosing the wetland $CH_4$ behaviour. This section briefly
describes those datasets used in the case study analysis of African wetlands in Sections 5.2 - 5.4.

### 5.1.1 Wetland Emissions Datasets

WetCHARTs (Bloom et al., 2017a) is a simple data-driven wetland model and one which previously been extensively evaluated
against satellite observations (Parker et al., 2020b). WetCHARTs is also commonly used as a priori information in atmospheric
inversions of $CH_4$ (Sheng et al., 2018; Lu et al., 2021; Palmer et al., 2021). As such, it can act as a useful benchmark against
which to compare the JULES wetland emission estimates.

We also utilise emission estimates from a dedicated high-resolution ($0.5° \times 0.625°$) atmospheric inversion of GOSAT $XCH_4$
(Lunt et al., 2021) using the GEOS-Chem model over sub-Saharan Africa. Emissions were estimated in a Bayesian inversion
framework between 2010 and 2016. Emission priors for wetlands were taken from the WetCHARTs model, EDGAR v4.3.2
database for anthropogenic emissions and the GFED v4.1s dataset for biomass burning emissions. Total $CH_4$ emissions were
resolved in the inversion from basis functions representing individual countries and major river basins. Posterior wetland
emissions were estimated based on the fraction of prior emissions from wetlands in each grid cell, scaled by the posterior total
$CH_4$ emissions.

### 5.1.2 Wetland Extent Datasets

Wetland extent information can either be obtained from prognostic (model-based) or observation-based estimates.
We use the Wetland Area and Dynamics for Methane Modeling (WAD2M) wetland extent dataset (Zhang et al., 2021) which
provides global 0.25° x 0.25° estimates of wetland fraction for inundated and non-inundated vegetated wetlands, derived from
microwave remote sensing. In this study we use the updated version which spans 2000-2018.

The global flood simulation model CaMa-Flood v4.0 (Yamazaki et al., 2011; Zhou et al., 2021) was used to predict fluvial in-
undation extents, specifically simulations at 0.25° x 0.25° resolution driven by JULES runoff estimates from the eartH2Observe
project (Marthews et al., 2021).

Finally, we also use surface reflectance imagery from the MODIS satellite, processed and visualised using Google Earth
Engine. This data allows a visual inspection of the region and provides a useful indicator of potential inundation, albeit not in
the presence of dense vegetation canopy.

### 5.1.3 Sentinel-5P TROPOMI $XCH_4$

We use $XCH_4$ from v1.5 of the University of Bremen TROPOMI WFMD retrieval (Schneising et al., 2019). Although the
TROPOMI data is relatively new (Sentinel 5-Precursor launched in October 2017) and algorithm development is still matur-
ing, TROPOMI does offer an unprecedented capability for mapping of $CH_4$ over large regions at an enhanced (7km) spatial
resolution and complements the long-time series of GOSAT point-based measurements.




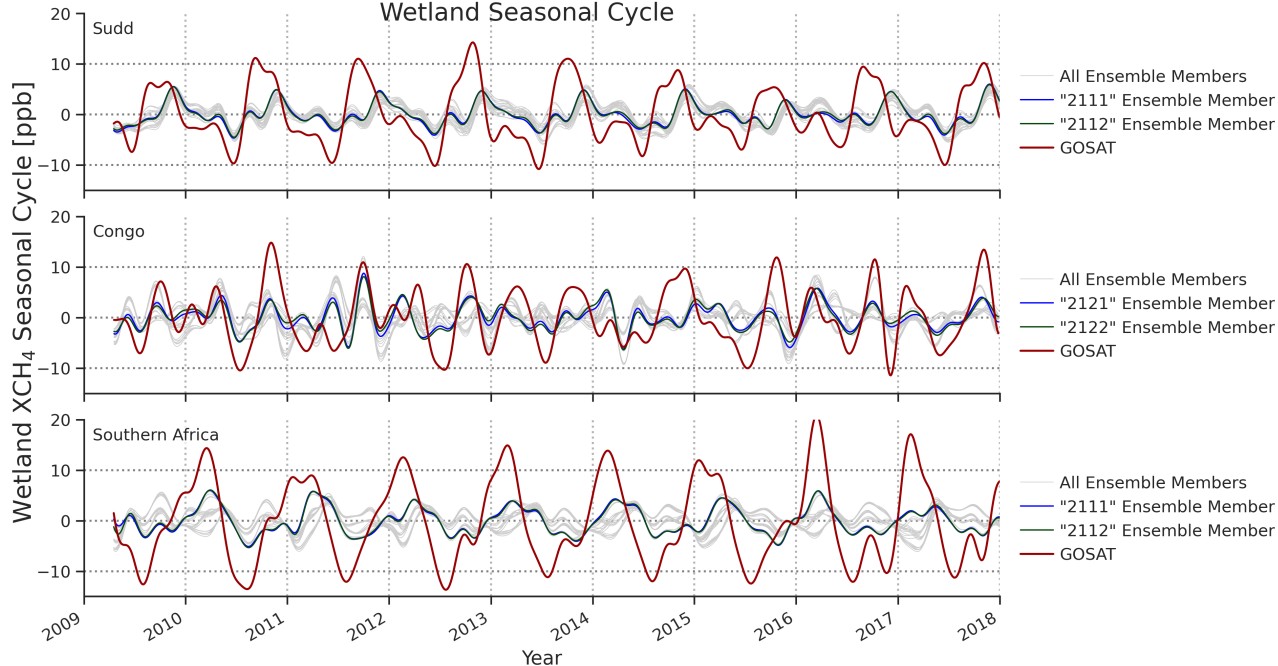

**Figure 8.** Time series showing the GOSAT (red) and JULES ensemble (grey) wetland CH$_4$ seasonal cycles for the three African wetland regions. The wetland seasonal cycle is calculated by subtracting the TOMCAT model simulations that do not contain any wetland emissions. For each region, the best performing (highest $R_{cycle}$) ensemble members are shown for the JULES and JULES-SWAMPS wetland extent configurations.

## 5.2 The Sudd

The first region we focus on is the Sudd wetlands in South Sudan. The Sudd is one of the world's largest freshwater ecosystems and the largest in the Nile Basin, draining much of Eastern Africa, including from Lake Victoria (Sutcliffe and Brown, 2018). This outflow from Lake Victoria leads to strong seasonal inundation, characterised by annual flood pulses (Rebelo et al., 2012), which is further modified by local precipitation and evaporation (Mohamed and Savenije, 2014), leading to highly complex and seasonal behaviour (Sosnowski et al., 2016). Previous work (Parker et al., 2020a; Lunt et al., 2021; Pandey et al., 2021) has

detailed the importance of understanding and characterising the CH$_4$ emissions from the Sudd wetlands given their sensitivity to large-scale climate drivers.

As discussed in Section 5, we find for all three African regions that neither parameterisation of wetland extent (JULES nor JULES masked with SWAMPS) outperforms the other and as shown in Figure 5, the correlation coefficient between the ensemble members and observations is poor. Although the WFDEI driving data greatly improves the correlation coefficient

compared to ERA-Interim, the best performing ensemble member are only capable of achieving an $R_{cycle}$ value of 0.61 (Figure A2). It is interesting to note here that for the Sudd, the ensemble members that perform best against observations


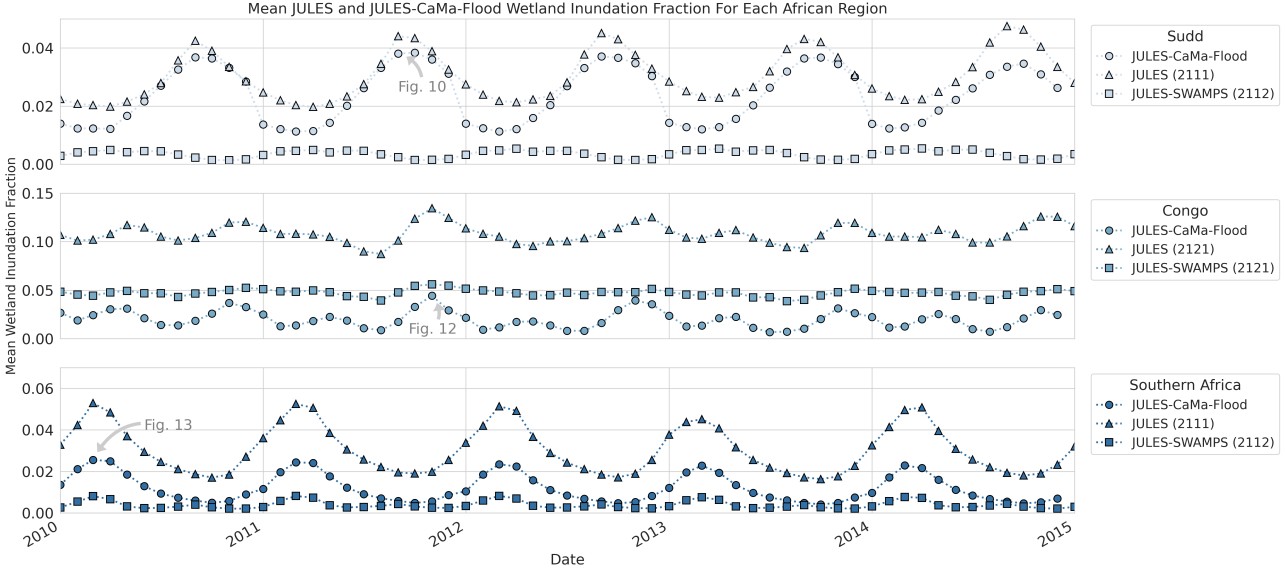

**Figure 9.** Time series showing the mean fluvial inundation fraction generated by the CaMa-Flood model for the three African wetlands regions between 2010 and 2015 compared to the standard JULES groundwater inundation. The annotations highlight the example month chosen for each region that are subsequently presented in Figures 10, 12 and 13.

(2121/2122: WFDEI, Phenology, high temperature dependency) are the exceptions from the ensemble. The majority of the ensemble members correlate well to each other and poorly to the observations. Figure 8 (top) shows the wetland seasonal cycle for the individual ensemble members and includes the observed seasonal cycle. The wetland seasonal cycle amplitude

($A_{JULES}$) even for the best performing ensemble member is significantly lower than the observed seasonal cycle ($A_{GOSAT}$) as summarised in Figure 5 and the reason for the poor $R_{cycle}$ value is that JULES appears to be out of phase with observations. This all suggests a fundamental lack of variability is being generated by JULES, with wetland extent an obvious parameter to evaluate in greater detail.

     We compare in Figure 9 the JULES wetland fraction for these three regions against that generated using JULES-CaMa-Flood

simulations which are capable of explicitly representing river and floodplain water dynamics and hence incorporate fluvial inundation. CaMa-Flood is the only open-source global river routing model that is based on the local inertial approximation of the Saint Venant equations, which takes into account the backwater and tide effects of downstream elements (viz. the possible reversal of flow in particular reaches upstream from e.g. lakes, tributaries, estuaries) (Marthews et al., 2021).

     For the Sudd we find that the wetland extent seasonal cycle and magnitude are very similar between JULES and JULES-

CaMa-Flood. However, applying the SWAMPS masking to JULES results in the JULES-SWAMPS configuration having a drastically smaller seasonal cycle amplitude and a significantly different phase (almost completely out of phase). This suggests that simply applying the JULES-SWAMPS mask for the Sudd results in a decoupling of the seasonal cycle for the masked areas from the wider region. For this reason, we evaluate the spatial distribution of both the $CH_4$ emissions and the wetland extent.





Figure 10 focuses on September 2011 which is towards the peak of the inundation as indicated by the JULES-CaMa-Flood
simulations (Figure 9). In Figure 10 we present $CH_4$ emission maps over the Sudd from two of the JULES ensemble members
(one with the default wetland extent (Fig. 10a) and one with the additional SWAMPS mask (Fig. 10b)). Furthermore, we also
show the $CH_4$ emissions derived from a GEOS-Chem flux inversion (Fig. 10e) and from the WetCHARTs ensemble (Fig. 10f).
In addition to the $CH_4$, we show the JULES wetland fraction (Fig. 10c), MODIS imagery (Fig. 10d), the JULES-CaMa-Flood
wetland fraction (Fig. 10g) and the WAD2M wetland fraction (Fig. 10h). By using this wide range of information we are able
to more confidently assess and evaluate the performance of JULES in this region and determine whether wetland area (and
subsequently $CH_4$ emissions) are being generated in the correct locations.

There is an obvious discrepancy between the area where JULES generates wetland area (and subsequently $CH_4$ emission)
compared to the location indicated by all of the other datasets. JULES places the majority of wetlands in the region in western
Ethiopa (Fig. 10c) and fails to generate significant wetlands in South Sudan. All of the other data sources agree strongly where
the wetlands and emissions should be located (Figs. 10d-h), the majority over the Al-Sudd wetlands in central South Sudan
with additional wetlands in the Machar marshes on the border with Ethiopia (e.g. Fig. 10h). When using the SWAMPS masking
of the JULES wetland extent, slightly more emissions are generated in the correct location due to the removal of the majority
of the spurious Ethiopian emissions but emissions remain significantly too small in both area and magnitude.

As further confirmation for where $CH_4$ emissions should be present in this region, $CH_4$ observations from TROPOMI are
used, allowing us to map $CH_4$ in the region. Figure 11 (bottom) shows the enhancement in the TROPOMI data over the Sudd
region, calculated by subtraction of latitudinal means, between January - May for 2018-2020. This clearly shows a strong
enhancement in the measured $CH_4$ total column (in excess of 45 ppb) at the location consistent with our above interpretation,
directly over the Al-Sudd wetlands as well as an enhancement over the Machar marshes. Pandey et al. (2021) have previously
shown a similar enhancement from TROPOMI over this region, adding further weight to our conclusions.
The reason that JULES fails to produce these wetlands is largely due to the topography in this region. Rainfall here occurs
in the Ethiopian Highlands, flowing downhill to maintain the Sudd wetlands. Without the addition of a river routing and
inundation mechanism within the JULES simulations, wetlands are instead created erroneously in the Ethiopian Highlands (as
indicated in Figure 10a).

It is important to highlight here that the JULES-CaMa-Flood simulations (Fig. 10g) are capable of producing wetlands in
the correct location and as such, future developments within JULES that incorporate some of the CaMa-Flood capabilities for
river routing and fluvial inundation would be expected to significantly improve the ability of JULES to successful reproduce
the correct temporal and spatial distribution of wetlands, and ultimately $CH_4$ emissions, over the Sudd region.

## 5.3 The Congo

The second region that we focus on is the Congo. The Congo Basin contains flooded forests and peatlands, known as the
Cuvette Centrale, which act as a major global store of carbon (Dargie et al., 2017) and source of $CH_4$ emissions (Borges et al.,
2015). $CH_4$ emissions from the Congo are still poorly constrained (Melton et al., 2013), with dense cloud-cover and forest
canopies making observations of both wetland extent (Salovaara et al., 2005; Bwangoy et al., 2010; Becker et al., 2018) and



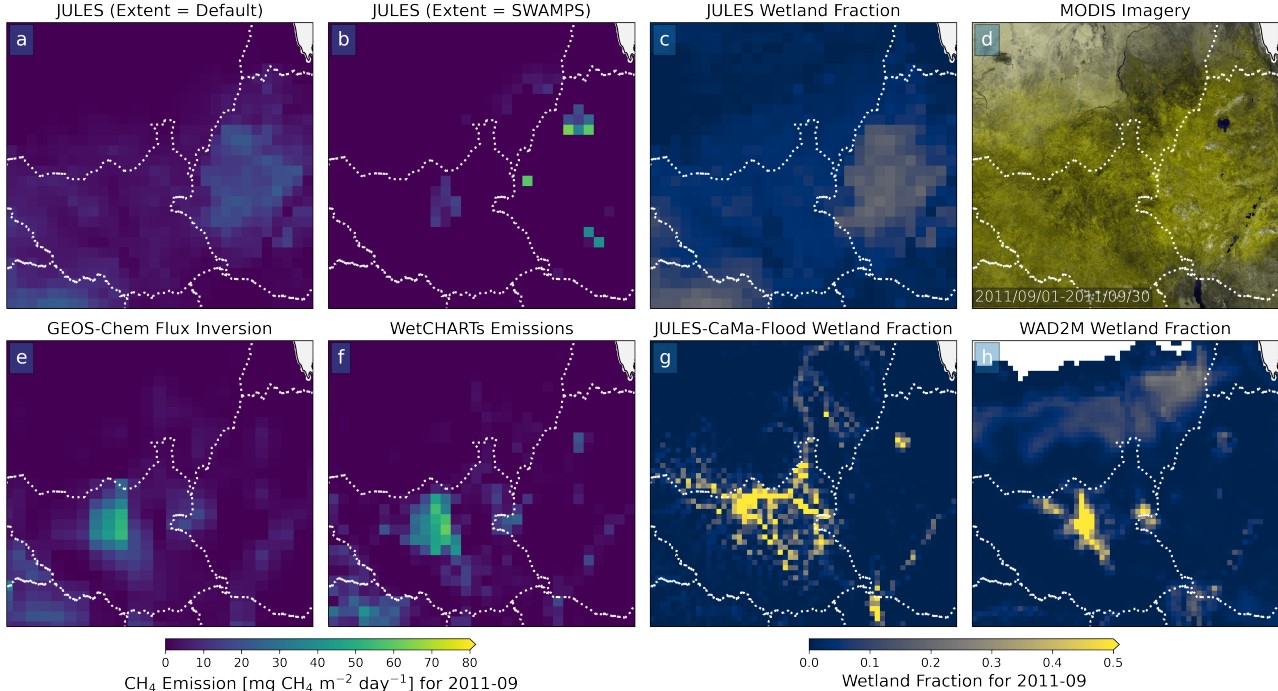

**Figure 10.** Comparison over the Sudd wetland region showing the wetland CH$_4$ emissions for September 2011 for a) JULES with the default wetland extent b) JULES with the SWAMPS masking for wetland extent d) GEOS-Chem flux inversion of GOSAT XCH$_4$ over Africa e) WetCHARTs ensemble mean. Also shown are the wetland fractions from c) JULES and f) CaMa-Flood. Both JULES simulations are the configurations that use the WFDEI meteorological driving, the lower $Q_{10}$ value and phenological vegetation as these were shown to provide the best result over this region (Fig: boxplot).

CH$_4$ emissions challenging (Tathy et al., 1992; Lunt et al., 2019; Parker et al., 2020b). The complex hydrology (Lee et al., 2011) in this region includes two wet seasons, in March and November (Haensler et al., 2013), making coupled climate simulations 400 of this region challenging (Crowhurst et al., 2021).

Figure 8 (middle) shows the modelled ensemble seasonal cycle along with the observed seasonal cycle. Again, the highest correlation coefficient for an ensemble member is found to be poor ($R_{cycle}$ = 0.52) with some ensemble members exhibiting zero correlation to the observations. This again suggests a significant lack of seasonal variability in the JULES simulations. Furthermore, the observed seasonality exhibits more complex behaviour with double-peaks in some (but not all) years, high-405 lighting the complex hydrology in this region.

Figure 9 (middle) shows that the seasonality produced by JULES-CaMa-Flood is in good agreement with that from JULES but with significantly lower average inundation $\sim 0.02$ vs $\sim 0.10$. When applying the SWAMPS masking to JULES, the average inundation is reduced (to $\sim 0.05$) with the seasonality is largely lost.


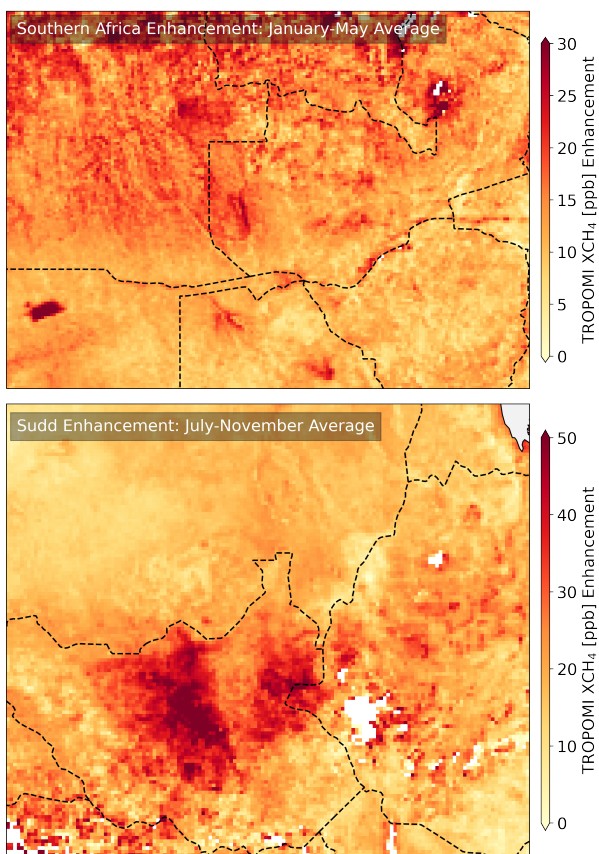

**Figure 11.** Enhancement in TROPOMI XCH$_4$ calculated by gridding the data into daily 0.1° x 0.1° bins and subtracting a baseline for each latitude bin. The baseline is calculated as the 5th percentile of each latitude bin with a rolling 5-bin (i.e. 0.5° latitude) average used to smooth out fluctuations. The enhancements are shown for the Southern Africa and Sudd regions, averaged over the months where the wetland signal peaks as indicated in Figure 8. It should be noted for Southern Africa that the enhancement over the Etosha Pan in Namibia (south-west corner of the domain) is likely overestimated due to particular spectral albedo variations within the fitting window used in the satellite retrievals. Finally, there were not sufficient cloud-free observations for the Congo region.

By comparing against the additional datasets we see why the Congo remains a difficult area to model. The default JULES
simulations lead to groundwater inundation of the entire Congo Basin (Fig. 12c), leading to fairly low widespread emissions whereas the JULES simulations with the extent masked by SWAMPS produce significantly more emissions (Fig. 12b), more tightly constrained to the area in the vicinity of the river system, albeit still very widespread. These latter emissions with the SWAMPS mask do appear to be in more reasonable agreement spatially with the CH$_4$ emissions from both the GEOS-Chem inversion (Fig. 12e) and from WetCHARTs (Fig. 12f). Some care needs to be taken here as WetCHARTs itself is used as the prior
for the GEOS-Chem flux inversion so the two should not be considered fully independent, and the major difference between them is reflected in the emissions magnitude. The fluvial inundation from the JULES-CaMa-Flood simulations over the Congo



(Fig. 12g) produce wetland extent close to the river which is largely missing from the standard JULES simulations. MODIS imagery (Fig. 12d) agrees with the JULES-CaMa-Flood simulations and does not show clear signs of inundation over this area except directly at the rivers. However, this may be misleading due to the dense tree canopy in this area. Indeed, wetlands (i.e. swamps and flooded forest) in the Congo can exist in relatively hilly areas, not directly fed by river flooding, but more due to local precipitation or groundwater. The pattern of wetland fraction from WAD2M (Fig. 12h), employing microwave observations that can partially penetrate the canopy layer, does suggest that there is a combination of both groundwater inundation and fluvial inundation. This does highlight the challenge in simulating such flooded forests where evaluation can be challenging and observations lacking. Additionally, dense cloud-cover in this region results in very few successful $CH_4$ retrievals from satellites (both GOSAT and TROPOMI), again reducing our capability to accurately evaluate model performance in this region.

The Congo remains one of the most significant global wetland regions but equally remains one of the most challenging to simulate and evaluate, with a significant uncertainty in the $CH_4$ emissions. Ongoing model development (Gedney et al., 2019) related to inclusion of methane emissions from trees in flooded areas (Pangala et al., 2017; Gauci et al., 2022) as well as improvements in the soil ancillary data to represent oxisol and ultisol soils in this area are expected to improve our ability to more accurately model the $CH_4$ emissions from the Congo in future work.

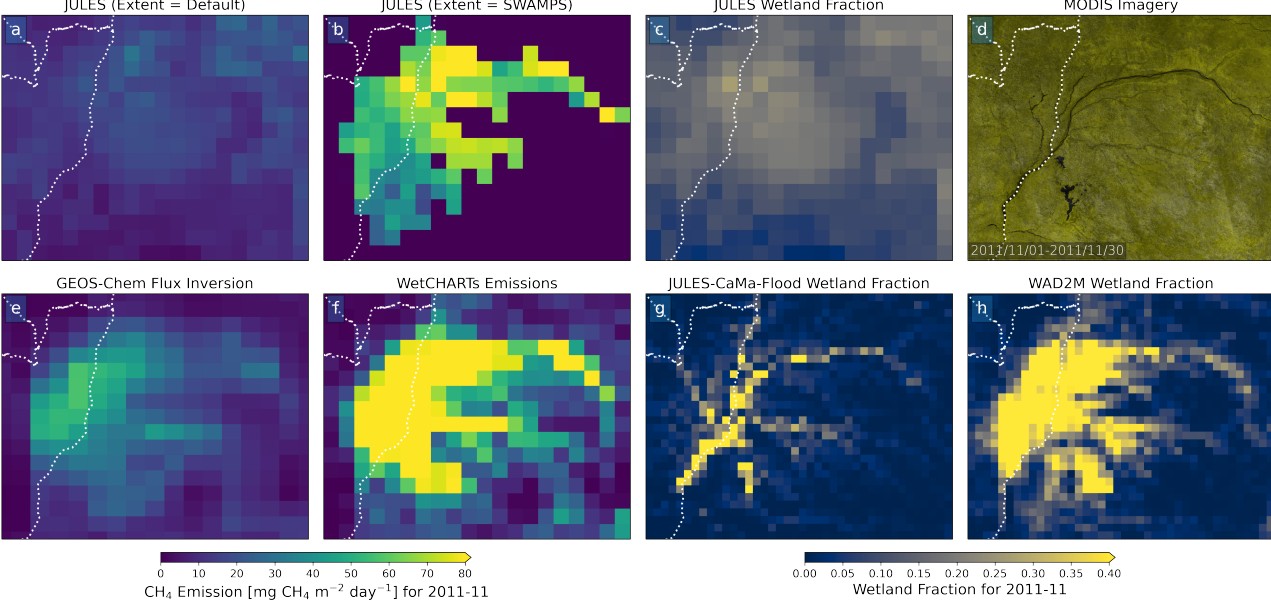

**Figure 12.** Comparison over the Congo wetland region showing the wetland $CH_4$ emissions for November 2011 for a) JULES with the default wetland extent b) JULES with the SWAMPS masking for wetland extent d) GEOS-Chem flux inversion of GOSAT $XCH_4$ over Africa e) WetCHARTs ensemble mean. Also shown are the wetland fractions from c) JULES and f) caMa-Flood. Both JULES simulations are the configurations that use the WFDEI meteorological driving, the higher $Q_{10}$ value and phenological vegetation as these were shown to provide the best result over this region (Fig: boxplot).

## 5.4 Southern Africa

The final region that we evaluate is Southern Africa, primarily focusing on the Zambezi River Basin in Zambia and Angola but also including parts of Namibia, Botswana, Zimbabwe, Mozambique and the Democratic Republic of Congo. Wetlands in this region are primarily swampland and seasonally inundated savannah/grasslands (Zimba et al., 2018; Lowman et al., 2018). The region also encompasses the Okavango Delta in northern Botswana (McCarthy, 2006; Wolski et al., 2012).


The values of $R_{cycle}$ for this region are found to vary significantly, ranging from reasonable positive correlations ($R_{cycle}$ = 0.67) to similar large negative correlations ($R_{cycle}$ = -0.68). This region is one in particular where the WFDEI-based ensemble members perform much better than the ERA-Interim members as shown in Figure 7.

Figure 8 (bottom) shows that for the ensemble members with the largest $R_{cycle}$ value, there is a reasonable correlation (maximum of 0.67) to the observed cycle. However, this is countered by some ensemble members having a similarly negative $R_{cycle}$ value (of -0.68 in the worst case). All of the ERA-Interim based ensemble members have a low or negative $R_{cycle}$ value (-0.68 - 0.23) whereas the WFDEI ensemble members range from -0.23 - 0.67. This very wide spread in $R_{cycle}$ (-0.68 - 0.67) across the ensemble explains why the average correlation is found to be very poor (Fig. 5).


When comparing the wetland extent from the best performing ensemble members to that produced by JULES-CaMa-Flood (Fig. 9 (bottom)) we find a good agreement in the seasonality between all three. However, in terms of the magnitude, the average groundwater inundation for the default JULES configuration is augmented by approximately 50% in the simulation with JULES-CaMa-Flood, with the SWAMPS-masked inundation in contrast being far too low. Figure 13 clarifies that although the seasonality is reasonable, the spatial distribution is again, incorrect. The default JULES wetland extent for this region places wetlands in northern Zambia and southern Democratic Republic of Congo. In contrast, the SWAMPS masking places the wetlands primarily along the Zambezi and Bangweulu wetlands in the west and north-east of Zambia respectively. The flux inversion results from GEOS-Chem suggest that emissions are observed over the Zambezi floodplain but also various other locations in the region including the Okavango Delta to the south, around Lake Kariba along the Zambia/Zimbabwe border, the Cahora Bassa lake in Mozambique and the Bangweulu wetland system in north-east Zambia. The WAD2M wetland fractions and the MODIS imagery both also indicate these as all being significantly inundated areas. Although the methane enhancement signals (20-30 ppb) are not as large as identified for the Sudd region, the TROPOMI S5P $CH_4$ enhancement (Fig. 11) does indicate enhanced $CH_4$ values over these areas giving further confidence that the inundated areas are being correctly identified along with their subsequent $CH_4$ emission by the GEOS-Chem flux inversion. The wetland fraction calculated by the JULES-CaMa-Flood simulation (Fig. 13g) is found to be in very good agreement with the WAD2M data (Fig. 13g) and hence suggests that JULES $CH_4$ emissions based on the JULES-CaMa-Flood derived wetlands would be in much closer agreement to the observations.





## 6 Conclusions

Overall we find that existing configurations of JULES can simulate wetland $CH_4$ emissions comparable in performance to those generated via state-of-the-art data-driven emission inventories such as WetCHARTs.





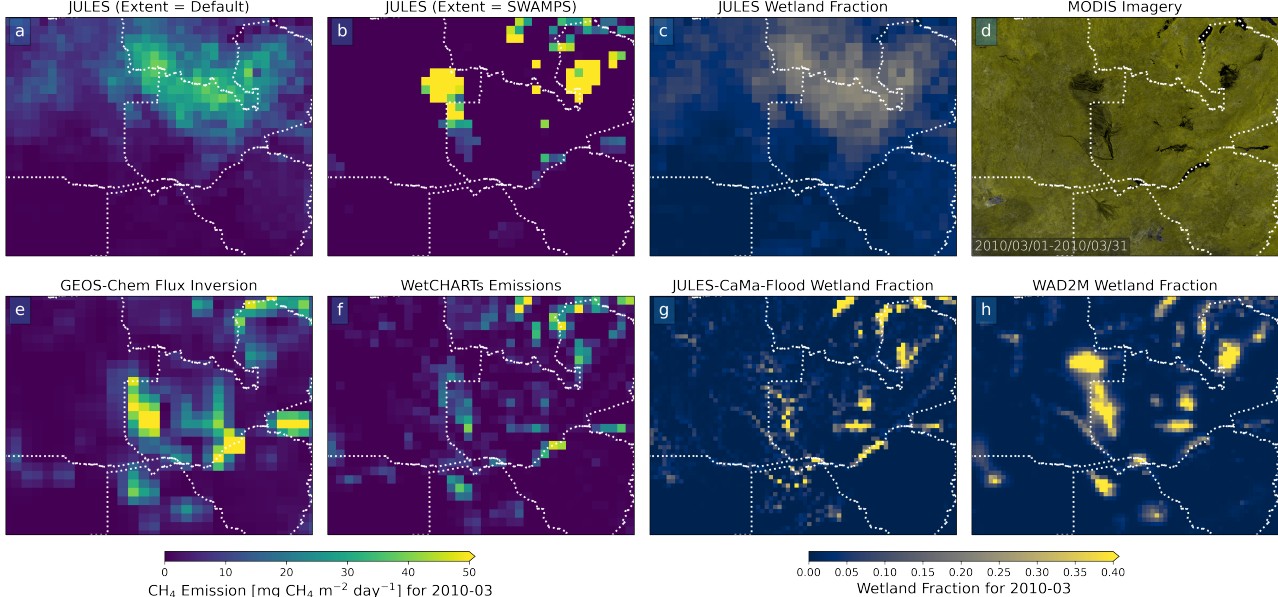

**Figure 13.** Comparison over the Southern Africa wetland region showing the wetland CH$_4$ emissions for March 2010 for a) JULES with the default wetland extent b) JULES with the SWAMPS masking for wetland extent d) GEOS-Chem flux inversion of GOSAT XCH$_4$ over Africa e) WetCHARTs ensemble mean. Also shown are the wetland fractions from c) JULES and f) caMa-Flood. Both JULES simulations are the configurations that use the WFDEI meteorological driving, the lower $Q_{10}$ value and phenological vegetation as these were shown to provide the best result over this region (Fig: boxplot).

The wetland methane seasonal cycle amplitude from JULES is typically underestimated compared to observations by be-
tween 1.8 ppb and 19.5 ppb across the different wetland regions examined. However, the correlation coefficient to the observed seasonal cycle is typically reasonable-to-good for most wetland regions (r = 0.58 to 0.88) although several regions do exhibit a poor correlation (r < 0.31) and these are explored in more detail.

Across the JULES ensemble, there are significant differences between ensemble members with the WFDEI driving data giving universally better performance than ERA-Interim. This highlights the vital role that the meteorological driving input
data has on determining the wetland response within the model and emphasises the benefits of bias-correcting to observations as done in the generation of the WFDEI data.

We find that the specific vegetation configuration of the ensemble member has a small effect on the performance (with Phenology typically performing better than either TRIFFID configuration) suggesting that there are potential improvements to consider when using a dynamic vegetation model such as TRIFFID. The effect of the temperature dependency is moderate,
with the lower value ($Q_{10}$ = 3.7) generally performing best but there are some important regional differences where the effect is much larger. We recommend further investigation into the variability in $Q_{10}$ across different ecosystems and the consequences that has for CH$_4$ emissions.





Neither choice of wetland extent, either the original JULES as is or masked with SWAMPS data, tends to perform better and both clearly have significant deficiencies. We find that a simple masking of the JULES wetland extent with the observed
SWAMPS wetland mask is not sufficient to reproduce the wetland seasonal cycle in key areas and instead, fundamental changes to the way the inundation is modelled are necessary in some regions, particularly those regions where fluvial inundation plays a significant role in the hydrology. This is demonstrated by the significant improvement in the agreement to multiple observation-based wetland and CH$_4$ datasets when using the JULES-CaMa-Flood wetland extent which incorporates fluvial inundation compared to the original (interfluvial) JULES data over key African wetland regions. Incorporating such fluvial
inundation changes into JULES is expected to significantly improve the ability of JULES to better represent the wetland extent and subsequently, produce more accurate CH$_4$ emissions.

Despite our analysis pointing towards the potential for significant improvements in key regions, the Congo wetland region in particular remains both challenging to model *and* to evaluate, highlighting the need for further study and additional ground-based observations that are less affected by the extensive cloud coverage of the region. Improved mapping of the wetland extent
(by both groundwater and fluvial inundation) as well as measurements of the temperature dependency of the CH$_4$ emissions would help in further constraining the CH$_4$ emissions from this region.

Finally, ongoing developments within JULES, such as the chimney venting of CH$_4$ by vegetation and the improved representation of soil properties, are expected to lead to additional improvements in the model. With these additions coupled to an improved representation of wetland extent and variability through more advanced hydrological modelling, we greatly improve
our capability to model the emission of CH$_4$ from tropical wetlands both historically and under a changing future climate.



*Code and data availability.* For this study, we use version 5.1 of JULES (at revision 10836, released in February 2018). The source code is available from the JULES code repository (see https://code.metoffice.gov.uk/trac/jules/log/main/trunk?rev=10836, user account required). The rose suites used for the specific JULES runs are: u-ba800 (WFDEI+phenology), u-bh665 (WFDEI+TRIFFID no competition), u-ax384 (WFDEI+TRIFFID), u-be476 (ERA Interim+phenology), u-be478 (ERA Interim+TRIFFID no competition) and u-be517 (ERA Interim+TRIFFID).

terim+TRIFFID). The rose suites can be found at https://code.metoffice.gov.uk/trac/roses-u/, (user account required). We run each rose suite twice, using $Q_{10}$ values of 3.7 and 5.0.

The latest version of the University of Leicester GOSAT Proxy v9.0 XCH$_4$ data (Parker and Boesch, 2020) is available from the Centre for Environmental Data Analysis data repository at https://doi.org/10.5285/18ef8247f52a4cb6a14013f8235cc1eb. The version used in this study (v7.2) is available from the Copernicus C3S Climate Data Store at https://cds.climate.copernicus.eu. WetCHARTs v1.0 is available from

Bloom et al. (2017b). This study uses v1.2.1 which is available on request from A. Bloom. WAD2M is available from https://doi.org/10.5281/zenodo.3998454. The MODIS Surface Reflectance 8-Day L3 data and MODIS Combined 16-Day NDWI data were visualised via the Google EarthEngine software with the data provided courtesy of the NASA EOSDIS Land Processes Distributed Active Archive Center (LP DAAC), USGS/Earth Resources Observation and Science (EROS) Center, Sioux Falls, South Dakota (https://lpdaac.usgs.gov). The University of Bremen TROPOMI/WFMD XCH$_4$ data are available from https://www.iup.uni-bremen.de/carbon_ghg/products/tropomi_wfmd/.

Requests for information about the code, data and parameterisations can be made to the corresponding author.

*Author contributions.* RJP generated the GOSAT XCH$_4$ retrievals, performed the analysis and drafted the manuscript. AAB produced an updated version of the WetCHARTs dataset for use in this study. CW and MPC produced the TOMCAT model simulations. GH and ECP generated the JULES simulations usage. TM generated the JULES-CaMa-Flood simulations. ML and PIP performed the GEOS-Chem flux inversion. All co-authors contributed to the planning and discussion of this study and on refining the manuscript.

*Competing interests.* We declare no knowledge of any competing interests.

*Acknowledgements.* RJP, HB, CW, MPC, PIP are funded via the UK National Centre for Earth Observation (NE/R016518/1 and NE/N018079/1). We acknowledge the support of the UK Natural Environment Research Council through the grants and awards: The Global Methane Budget (MOYA, NE/N015681/1, NE/N015657/1 and NE/N015746/1), The UK Earth System Modelling project (NE/N017951/1) and Hydro-JULES (NE/S017380/1). Part of this research was carried out at the Jet Propulsion Laboratory, California Institute of Technology, under a contract

with the National Aeronautics and Space Administration. Funding for the WetCHARTs emissions was provided through a NASA Carbon Monitoring System Grant NNH14ZDA001N-CMS. We also acknowledge funding from the ESA GHG-CCI and Copernicus C3S projects.

This research used the ALICE High Performance Computing Facility at the University of Leicester for the GOSAT retrievals and analysis. We undertook the JULES runs on the NERC's JASMIN High Performance Computing Facility. The TOMCAT simulations were performed on the national Archer and Leeds Arc HPC systems.

This publication contains modified Copernicus Sentinel data (2018,2019,2020). Sentinel-5 Precursor is an ESA mission implemented on behalf of the European Commission. The TROPOMI payload is a joint development by the ESA and the Netherlands Space Office (NSO).





Sentinel-5 Precursor ground-segment development has been funded by the ESA and with national contributions from the Netherlands, Germany, and Belgium. The generation of the TROPOMI methane product by University of Bremen has been funded by ESA (GHG-CCI+ project) and by the State and the University of Bremen

We thank the Japanese Aerospace Exploration Agency, National Institute for Environmental Studies, and the Ministry of Environment for the GOSAT data and their continuous support as part of the Joint Research Agreement.

**Appendix A**

In the main text (Section 4) we make reference to previous work (Parker et al., 2020b) we have undertaken to evaluate the WetCHARTs data-driven emission inventory (Bloom et al., 2017a) using a similar methodology as used in this study. That

allows a direct comparison between the performance of the JULES wetland $CH_4$ emissions for these regions to the WetCHARTs performance. Figure A1 reproduces Figure 5 from this study and compares to Figure 4 from Parker et al. (2020b).

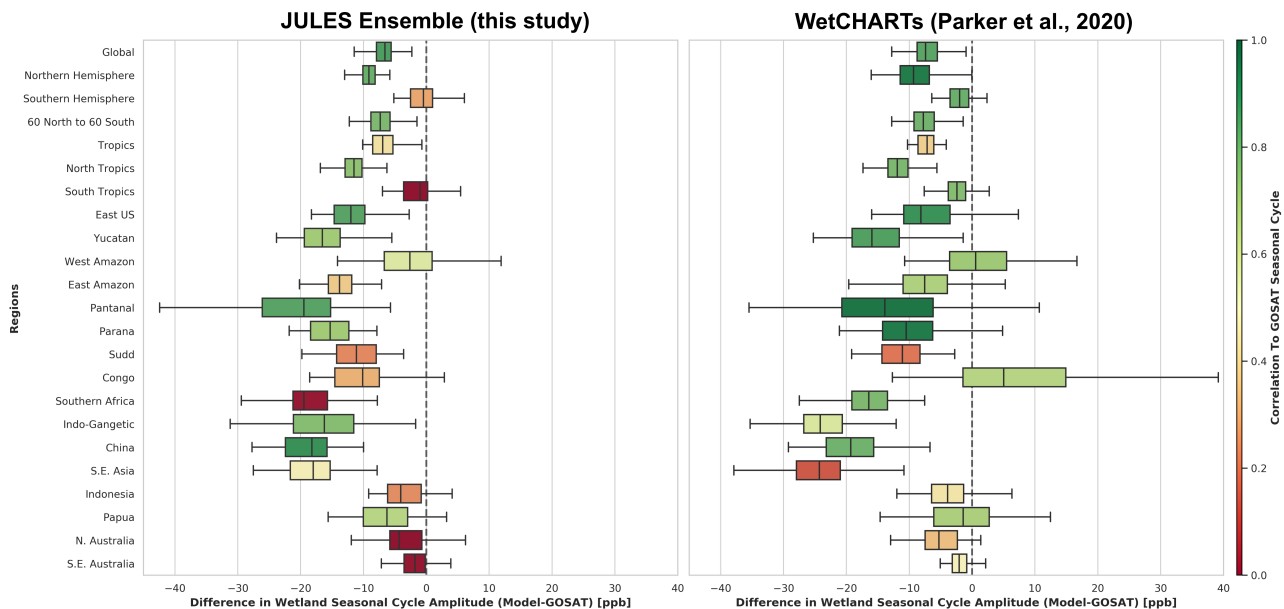

**Figure A1.** Comparison of Figure 5 from this study for JULES against Figure 4 from Parker et al. (2020b) for WetCHARTs.

Figure A2 shows the correlation coefficient between the different ensemble members and the observed wetland $CH_4$ seasonal cycle for the Sudd region. The majority of the ensemble members correlate strongly to each other (r > 0.9) but poorly to the observed seasonal cycle (r < 0.2). The set of ensemble members that correlate best to observations (members 2121 and 2122

- WFDEI meteorology, Phenology vegetation and high $Q_{10}$ value) correlates the least to the remaining ensemble members, suggesting a significant difference in the characteristics of these few ensemble members. This is discussed in the main text in Section 5.2.



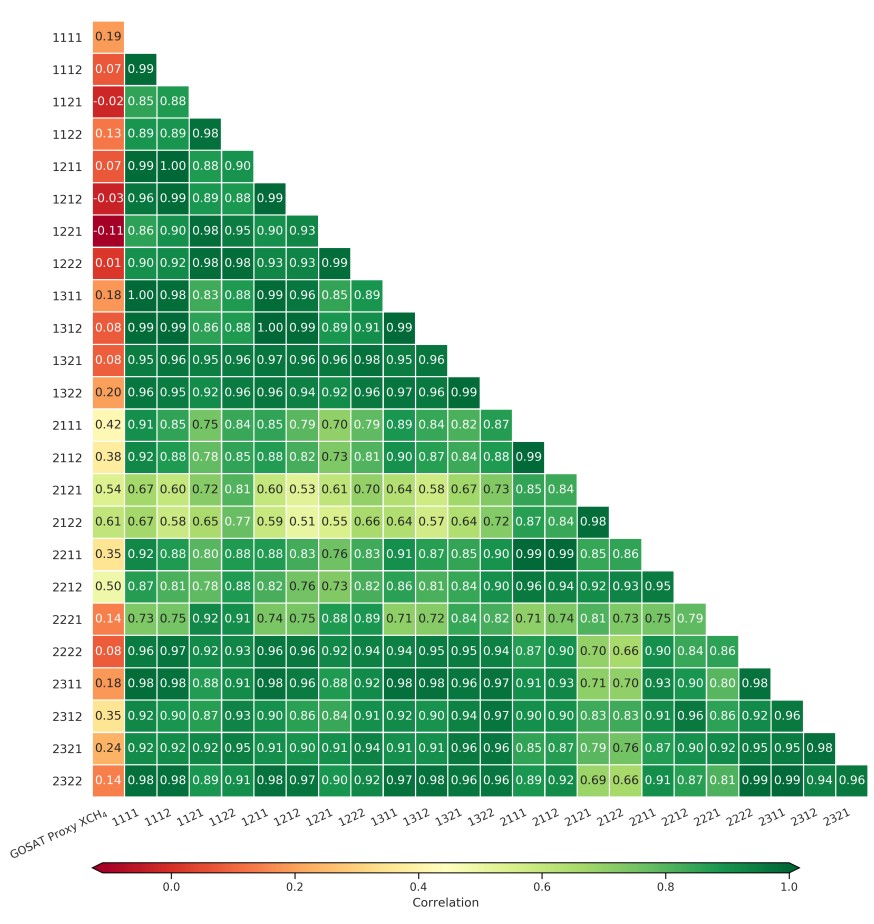

**Figure A2.** Correlation coefficient between the different ensemble members and the observed wetland CH$_4$ seasonal cycle for the Sudd wetland region.



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
