# Peer review of "Evaluation of Wetland CH4 in the JULES Land Surface Model Using Satellite Observations"

_Biogeosciences, 2022_

## Author Response (AR1)

**Author Comments**

**Colour Key:** Reviewer Comment     Our Response     New Manuscript Text

**Reviewer 1**

We thank the reviewer for their comments and appreciate them taking the time to review our study.

From how I read the paper, there is a dependence upon accurate anthropogenic/other natural/fire CH4 emissions for the attribution to wetlands from the GOSAT/TOMCAT retrievals. It appeared to me that those non-wetland sources were assumed to be perfect (along with the atmospheric inversions). I would have liked to see some attempt to understand how reasonable these other CH$_4$ source estimates were as all error terms were then pushed into the wetland methane emissions.

We agree with the comment (from both reviewers) that the non-wetland CH$_4$ emissions have the potential to cause issues with our analysis. We have attempted to mitigate these issues in the following ways:

Firstly, we deliberately maintained the same setup as used in Parker et al. (2020) so that we could directly compare the evaluation of JULES to that performed for WetCHARTs. This consistency allows us to draw wider conclusions about the utility of JULES compared against the state-of-the-art data-driven WetCHARTs dataset.

Secondly, the TOMCAT (non-wetland) model setup in this study (and Parker et al. 2020) uses a configuration of the TOMCAT model that has successfully been used in a range of studies. The model set-up, or a very similar one, has been used previously in McNorton et al. (2016), Wilson et al. (2016), Parker et al. (2018) and Parker et al. (2020). The emissions used in the current paper were also used as priors in CH$_4$ inversions documented in McNorton et al., (2018), Wilson et al. (2021) and Gloor et al. (2021).

Although the uncertainty in wetland methane emissions is typically larger than that from biomass burning, we acknowledge that in some regions the uncertainty on biomass burning may not be insignificant. We have performed (Wilson et al., 2021) a full global inversion of CH$_4$ flux using the same GOSAT data and model configuration as used in this study. Our findings from that study suggest that flux estimates in fire-affected regions (e.g. in South America and Africa) are generally consistent with the prior values. Despite wetland and burning regions often being spatially (and temporally) distinct, there is the possibility of some interference although any effect is expected to be small. In future work we plan to use newly-developed CO inversions to better represent the CH$_4$ flux from biomass burning.

The uncertainty introduced through model atmospheric transport and chemistry errors are likely larger than those from non-wetland emissions, and could be quantified in future analysis through use of multiple atmospheric transport models and/or representations of

transport and chemistry. However, the TOMCAT model has been shown in many studies to represent atmospheric transport of $CH_4$ very well (see references above).

In our revised manuscript we will follow the suggestion from Reviewer 1 and provide a more detailed discussion on the impact of these assumptions:

We do make the assumption that the uncertainties in the inter-annual variability of non-wetland $CH_4$ sources (such as biomass burning) are much smaller than the uncertainty in wetland methane emissions. This assumption has previously been tested (e.g. Parker et al., 2020, Wilson et al., 2021) and inversion results suggest that whilst it is possible for fire emissions to interfere with our analysis to a small degree, this is largely not the case with flux changes in fire-affected regions generally remaining consistent with the prior. In future work, CO inversions, currently under development, will allow us to better represent the $CH_4$ flux from biomass burning and separate any effect more explicitly.

In order to show the model performance in non-wetland regions, we will add this additional analysis (as suggested by Reviewer 2) into the appendix as a diagnosis of the non-wetland simulations.

We have performed analysis over three non-wetland areas (as highlighted in red in Figure 1), namely West US, Arabian Peninsula and Western Australia. These regions would not be expected to be dominated by wetland emissions and hence evaluation of the simulated $CH_4$ column against observations provides an assessment of how the non-wetland emissions in the model are performing. The detrended methane seasonal cycle for the model is compared against GOSAT observations in Figure 2 and we find a very good agreement (with correlation coefficients of 0.89, 0.96 and 0.90, respectively, for West US, Arabian Peninsula and Western Australia).

[Figure]

*Figure 1 - As Figure 3 in main text but including 3 non-wetland background regions highlighted in red.*

[Figure]

*Figure 2: Comparison of the modelled methane seasonal cycle to GOSAT observations over the three background regions.*

Minor comments:

- Line 78: 'A deep layer of restrictive water flow' - does that just mean that you provide a no flow condition at 3 m?

Yes, that's correct. JULES appends a 'bedrock layer' below the simulated soil layers and water cannot move into the bedrock (and with the standard configuration the soil layers stop at 3 m).

- L109: Why is the time series scaled to 180 in particular? Why is this step necessary or desired?

The emissions from JULES are scaled so that the global total wetland methane annual emissions for 2000 has a value of 180 Tg/yr to ensure consistency with the best estimate of this value from Saunois et al. (2016). This is the same approach as used for JULES simulations in Comyn-Platt et al. (2018). If this step were not performed, the geographical masking of the JULES emissions by the SWAMPS wetland mask would result in low wetland emissions as rather than re-apportioning the fluxes to the wetland area, they would simply be discarded. This global re-scaling is a common approach and also utilised in the WetCHARTs model (Bloom et al., 2017).

We have clarified the reason for this in the manuscript.

In a post-processing step, the time series of annual wetland emissions of each ensemble member is separately scaled to give annual emissions of 180 Tg $CH_4$ $yr^{-1}$ for the year 2000 (Saunois et al., 2016), as described in Comyn-Platt et al. (2018). This step is necessary as the geographic masking of the JULES wetland area with the SWAMPS data would otherwise result in unrealistically low methane emissions due to the more limited geographic area.

- Fig 2: Do all of those in the grid actually give 180 Tg/yr in 2000? Ones like the bottom left seem to hardly be able to (although I realize the time shown is Aug 2011)

Yes, the global total for 2000 is scaled to 180 Tg $yr^{-1}$ so as to be consistent with Saunois et al. (2016).

- L 137: When a single C pool is used does that mean both the litter and soil (humified) C are tracked in only one pool?

For soil_bgc_model = 1 the single carbon pool is prescribed and not updated in time. We will update the manuscript to clarify this.

..single (fixed) soil carbon....

- L 150: Why use the SWAMPS dataset by itself, with its known inability to detect saturated, but not inundated, wetlands, and not make use of something like WAD2M? I see you use WAD2M later so are definitely aware of it.

Part of this was a practical reason. At the time when we ran our simulations the WAD2M data were not available. These data became available whilst we were performing our final analysis and hence for completeness we incorporated it into the study as part of the interpretation of the results. Reproducing the JULES data with the WAD2M data would be a significant undertaking and hence more suited to a future study, especially given the promising results from CaMa-Flood which we would also incorporate in future work.

Further, we do not expect our use of SWAMPS data instead of WAD2M data to affect the main arguments reported in the study, as ultimately we are evaluating the performance of the JULES land surface model. In either case, the application of an external wetland mask of the simulations is non-ideal and instead the focus is on identifying the processes required in future versions of JULES (such as fluvial inundation) that would allow accurate wetland simulation and negate the requirement for such masking. The use of WAD2M (along with WetCHARTs, GEOS-Chem flux inversions, JULES-CaMa-Flood and MODIS Imagery) in the analysis allowed a strong narrative to be constructed towards these goals.

- line 179 - fix ref.

Fixed

- Fig 4 - what are the units?

The units are in ppb as indicated in the y-axis title. We will make this clear in the figure caption.

- L 320 - WAD2M uses more than microwave remote sensing. Perhaps give a bit more detail here otherwise it sounds like it is just SWAMPS (which does form the seasonality but there are other important differences)

We have updated Section 5.1.2 to provide additional details on WAD2M to address this.

We use the Wetland Area and Dynamics for Methane Modeling (WAD2M) wetland extent dataset (Zhang et al., 2021) which provides global 0.25° x 0.25° estimates of wetland fraction for inundated and non-inundated vegetated wetlands. WAD2M is derived using a combination of surface inundation based on microwave remote sensing data along with static datasets that identify inland waters, agricultural areas, shorelines, and non-inundated wetlands. Areas containing permanent water bodies (such as lakes, rivers, etc), rice paddies and coast wetlands are excluded. The resulting dataset therefore represents the spatiotemporal patterns of inundated and non-inundated vegetated wetlands and is expected to improve estimates of wetland $CH_4$ fluxes. In this study we use the updated version which spans 2000-2018.

- Fig 12 - missing reference at end? (Fig: boxplot)?

Fixed

- Line 492 - chimney venting? Is this aerenchymal transport that is meant?

Yes. We have clarified this in the text.

Finally, ongoing developments within JULES, such as the "chimney venting" of $CH_4$ by vegetation (i.e. aerenchymal transport)...

- Code availability - user account required limits reviewers ability to check over code (should they wish to remain anonymous).

We absolutely agree with this and it is part of a wider issue with access to code such as JULES. There is an ongoing initiative to relax the license and make the JULES code openly available. Unfortunately that is not yet available and as such, as a compromise, access is available via a user account.

- L 532- doesn't quite make sense. Needs rewording.

We have reworded this sentence so it is clearer.

**References**

Bloom, A. A., et al., A global wetland methane emissions and uncertainty dataset for atmospheric chemical transport models (WetCHARTs version 1.0), Geosci. Model Dev., 10, 2141–2156, https://doi.org/10.5194/gmd-10-2141-2017, 2017

Comyn-Platt, E. et al, Carbon Budgets for 1.5 and 2 °C Targets Lowered by Natural Wetland and Permafrost Feedbacks, Nature Geoscience, 11, 568–573, https://doi.org/10.1038/s41561-018-0174-9, 2018

Gloor M, et al., Large Methane Emissions From the Pantanal During Rising Water-Levels Revealed by Regularly Measured Lower Troposphere $CH_4$ Profiles. *Global Biogeochemical Cycles.* **35**(10) https://doi.org/10.1029/2021GB006964, 2021.

McNorton J., et al., Role of regional wetland emissions in atmospheric methane variability. *Geophysical Research Letters.* **43**(21), pp. 11433-11444, https://doi.org/10.1002/2016GL070649, 2016.

McNorton, J., et al., Attribution of recent increases in atmospheric methane through 3-D inverse modelling, Atmos. Chem. Phys., 18, 18149–18168, https://doi.org/10.5194/acp-18-18149-2018, 2018.

Parker, R. J. et al., Evaluating year-to-year anomalies in tropical wetland methane emissions using satellite CH4 observations, Remote Sensing of Environment, https://doi.org/10.1016/j.rse.2018.02.011, 2018

Parker, R. J. et al., Exploring constraints on a wetland methane emission ensemble (WetCHARTs) using GOSAT observations, Biogeosciences, 17, 5669–5691, https://doi.org/10.5194/bg-17-5669-2020, 2020.

Saunois, M. et al., The global methane budget 2000–2012, Earth Syst. Sci. Data, 8, 697–751, https://doi.org/10.5194/essd-8-697-2016, 2016.

Wilson C., et al., Contribution of regional sources to atmospheric methane over the Amazon Basin in 2010 and 2011. *Global Biogeochemical Cycles.* **30**(3), pp. 400-420, https://doi.org/10.1002/2015GB005300, 2016.

Wilson, C., et al., Large and increasing methane emissions from eastern Amazonia derived from satellite data, 2010–2018, Atmos. Chem. Phys., 21, 10643–10669, https://doi.org/10.5194/acp-21-10643-2021, 2021.

**Author Comments**

**Colour Key:** Reviewer Comment     Our Response     New Manuscript Text

**Reviewer 2**

We thank the reviewer for their comments and appreciate them taking the time to review our study.

- The TOMCAT model is used to bridge between methane emissions and GOSAT observed column averaged mixing ratios. A model run without wetland emissions serves as a reference that is subtracted from the GOSAT data to derive an 'observational' dataset that is used to evaluate different configurations of the wetland model. It should be made clearer that this evaluation depends critically on the validity of the TOMCAT simulation without wetlands. The uncertainty of that simulation should receive more attention. The implicit assumption is that this uncertainty is small compared to uncertainties due to wetland emissions. However, no evidence is presented in its support. It would have been easy to include a figure comparing TOMCAT to background measurements and assess whether the model – data mismatch is consistent with wetlands as the most uncertain component. It is true that some important other sources do not show a strong seasonality, but due to seasonal variations in atmospheric transport their impact on total column methane will nevertheless vary seasonally.

Please see our response to Reviewer 1 who makes a similar point. We agree with the reviewer that including some additional analysis into the manuscript would strengthen the arguments that we are making in relation to the reliability of identifying an observed wetland methane signal.

- Even if the model performs well against background measurements, this is not a guarantee that GOSAT – model differences are due to wetland emissions. This needs to be acknowledged somewhere.

Please see our detailed response to Reviewer 1 who makes a similar point.

- An ensemble of wetland configurations is used to represent the uncertainty of wetlands, including an alternative representation of meteorology. However, it is unclear why the alternative meteorology is only used to drive the emission computation and not its transport in the atmosphere.

The purpose of this study was to evaluate the capability of the JULES land surface model to reproduce observed wetland methane emissions. Although in order to perform the evaluation using satellite data, these wetland fluxes need coupling with an atmospheric chemistry transport model, it is not within scope of this study to evaluate the performance of the model atmospheric transport itself. We will however make this clearer in the manuscript text as the reviewer is

correct that it does introduce an additional uncertainty in the analysis which should be acknowledged.

- It would have been useful to include another representation of the global methane sink and methane sources other than wetlands in the ensemble.

We were very conscious in this study that the purpose is to evaluate JULES and it definitely was not to look into wider issues relating to global sources/sinks which are beyond the scope of this work. The Global Methane Budget team (e.g. Saunois et al., 2020) does an exceptional job at this and both JULES and GOSAT CH4 data are incorporated into that analysis.

- It is unclear how the TOMCAT model tracers are initialized. If the model starts at 2009, when also the comparison with GOSAT starts, the initialization needs to be very good to do without a spin-up to bring the global methane source and sink for each tracer in balance. An explanation is needed of how this was done.

We will add these additional details in to the manuscript text to clarify how the spin-up was performed:

The model was initialised using the same method as Parker et al. (2018) and Parker et al. (2020), which in turn were based on simulations from McNorton et al. (2016). The model tracers were initialised in 1977 and run up to 2004 at coarser resolution (2.8°) than the main simulation. At this point the tracers were scaled to match the overall observed surface concentration for CH4. The period 2004 - 2009 was then run at the 1° resolution, before the analysis begins in 2009.

- Figure 5: It is unclear why the correlation color legend starts at zero. How would negative correlations show up?

These are the correlation values per region:

| Global | 0.85 |
|---|---|
| Northern Hemisphere | 0.81 |
| Southern Hemisphere | 0.29 |
| 60 North to 60 South | 0.77 |
| Tropics | 0.43 |
| North Tropics | 0.73 |
| South Tropics | 0.00 |
| East US | 0.83 |

| | |
|---|---|
| **Yucatan** | 0.71 |
| **West Amazon** | 0.58 |
| **East Amazon** | 0.37 |
| **Pantanal** | 0.83 |
| **Parana** | 0.70 |
| **Sudd** | 0.23 |
| **Congo** | 0.31 |
| **Southern Africa** | 0.01 |
| **Indo-Gangetic** | 0.76 |
| **China** | 0.88 |
| **S.E. Asia** | 0.46 |
| **Indonesia** | 0.24 |
| **Papua** | 0.69 |
| **N. Australia** | -0.16 |
| **S.E. Australia** | 0.01 |

*Table 1 - Correlation coefficients as described and presented in Figure 5 in the manuscript.*

We limited the colour scale to start from 0 as only Northern Australia showed a negative correlation and as can be seen by Figure 5 in the manuscript, the wetland signal in Northern Australia is very small (<5 ppb) and hence a correlation is not particularly meaningful. We will however clarify this in the text manuscript.

The lower-limit of the colour scale in Figure 5 is capped at 0, although it should be noted that one region (N. Australia) has a negative correlation of -0.16. However, the seasonal cycle over this region is very small (<5 ppb) and hence the correlation is not particularly meaningful.

- How are regional averages in Figure 5 taken? I suppose that model has been sampled to the coordinates of the GOSAT soundings? But then the global and other region averages are weighted by the uneven coverage of the GOSAT data. In addition, the impact of regional emissions on the total column is not limited to the region where the emission takes place. It could be that emissions from another region contribute more to the reported variability of methane over a region that the sources that are located there. How is this issue dealt with?

Yes, the model is sampled at the time/location of the GOSAT measurement with the scene-specific averaging kernels applied. The time series is then produced by averaging this data over the region and the statistics for the wetland seasonal cycle are produced as outlined in Section 4.

While it is true that some of the total column signal observed by GOSAT will be influenced by emissions from outside of the region considered, this is much less of an issue for GOSAT than it would be for other instruments. GOSAT measurements measure in the shortwave infrared and a consequence of this is that the observations are particularly sensitive to the surface (and hence to emissions). This is in contrast to a thermal infrared instrument (such as IASI) which is most sensitive to the mid-troposphere.

- Related to the previous point: what could be the influence of the seasonally varying coverage of the GOSAT measurements on the derived seasonality for a particular region? How do you avoid that spatial differences between GOSAT and TOMCAT "alias" into apparent seasonal differences?

As we were focusing over tropical wetlands, the seasonal coverage is not as extreme as it would be at high latitude. The reviewer is correct, were we examining boreal wetlands this would be a much larger issue. Furthermore, as our GOSAT retrieval uses the "proxy" method, we are far less susceptible to cloud cover interfering with the retrieval or introducing seasonal sampling biases. GOSAT measures in a gridded pattern, returning to the same location each time and hence the spatial coverage remains consistent.

The figure below (Figure 3) shows an example of this spatial coverage in the form of the sampling density for the African continent, highlighting the very regular measurement locations that are routinely returned to. As discussed in the conclusions to the manuscript text, the Congo area remains challenging to evaluate fully due to the difficulty in obtaining observations and although the observations that are obtained are consistently sampled over space and time, they are much fewer of them than in the other areas we examine.

[Figure]

*Figure 3: Figure showing the GOSAT measurement density over Africa. The three African wetland regions that we study (Sudd, Congo, Southern Africa) are highlighted by the black boxes.*

- Figure 6: Why are changes in correlation coefficient and standard deviation only in positive direction? What happens if the correlation coefficient or standard deviation of the subset is less? If these plots represent the absolute value of changes than the explanation in the caption about improvement or worsening makes no sense. An extended explanation is needed here.

We have attempted to explain this in the manuscript but we will modify this to present a clearer clarification. The values we report are the **change above the minimum value** for each pair/triplet. By construction this is a positive (or zero) value.

For clarity, the values that we report are the change above the minimum value for each pair/triplet and hence, by construction, this is a positive (or zero) value.

- Given its importance, it would be useful – without much work – to differentiate the impact of meteorology further. Is precipitation the dominant factor?

By choosing two different sets of meteorological data (ERA-Interim vs WFDEI) we have attempted to show this effect already. An analysis of the specific differences between these datasets is beyond the scope of this work and is already covered elsewhere (Weedon et al., 2014) but we do highlight some key differences in the manuscript and we will adjust the text to make these more explicit.

In this context, an important factor is that WFDEI precipitation is bias corrected using the observed monthly mean (Weedon et al., 2014). This is likely the cause of the significant improvement in wetland extent obtained by using WFDEI over ERA-Interim.

- Figure 8: The two ensemble member need more distinct colors to be able to see which is which.

We will adjust the colours (currently red and green) to make them more distinct.

- Figure 10, 12 and 13: the references to subfigures in the captions is wrong. What is the color legend of the MODIS imagery, is this RGB?

Fixed captions. The MODIS imagery is RGB and we will clarify this in the captions.

MODIS Imagery (RGB of surface reflectance)

**References**

McNorton J., et al., Role of regional wetland emissions in atmospheric methane variability. *Geophysical Research Letters.* **43**(21), pp. 11433-11444, https://doi.org/10.1002/2016GL070649, 2016.

Saunois M. et al., The Global Methane Budget 2000–2017, Earth Syst. Sci. Data, 12, 1561–1623, https://doi.org/10.5194/essd-12-1561-2020, 2020.

Parker, R. J. et al., Evaluating year-to-year anomalies in tropical wetland methane emissions using satellite CH4 observations, Remote Sensing of Environment, https://doi.org/10.1016/j.rse.2018.02.011, 2018

Parker, R. J. et al., Exploring constraints on a wetland methane emission ensemble (WetCHARTs) using GOSAT observations, Biogeosciences, 17, 5669–5691, https://doi.org/10.5194/bg-17-5669-2020, 2020.

Weedon, G. P., et al, The WFDEI Meteorological Forcing Data Set: WATCH Forcing Data Methodology Applied to ERA-Interim Reanalysis Data, Water Resources Research, 50, 7505–7514, 775 https://doi.org/10.1002/2014WR015638, 2014

---

## Referee Report (RR1)

**Referee report of revised BG manuscript:**
„Evaluation of Wetland $CH_4$ in the JULES Land Surface Model Using Satellite Observations"
by Robert J. Parker et al.

In this manuscript it is presented an evaluation of modeled wetland $CH_4$ emissions using the JULES Land Surface Model against atmospheric $CH_4$ concentrations from GOSAT satellite observations. For this comparison, the surface $CH_4$ emissions produced with ensemble simulations by JULES, are used in the global chemistry transport model TOMCAT in order to remove the non-wetland methane emissions from the modeled methane emissions.
A total of 24 ensemble members were produced using JULES, and those ensembles are a combination of varying atmospheric forcing data, vegetation, temperature dependence of soil-producing methane and wetland extent configuration.

The simulation results are at global scale; however, this work focuses mostly in tropical areas. This work provides a step forward in the evaluation of modeled $CH_4$ emissions in tropical wetland areas, a region of great interest in the global methane budget and of challenge for models. The study also offers a methodological consistent approach (ensemble) that was followed to analyze the performance of a land surface model under different criteria, providing key findings for other model studies.

The performance of the modeled emissions by the land surface model is considerably improved by including the SWAMP wetland extent. This approach was used to improve/deal with wetland extent in land surface models and brings an additional step forward to a known issue in the $CH_4$-wetlands modeling community: modelling wetland extent. It would be interesting to evaluate such performance in the more challenging boreal regions.

The original manuscript was reviewed by two referees who provided comments that were well responded by the authors. After reading the author comments, I recommend this work to be published in Biogeosciences.

Below, I report a summary of the general points discussed in the revision of this manuscript and provide own minor comments in the hope they can further support the improvement of this manuscript during the peer review process:

**1) The considerations taken for non-wetland $CH_4$ emissions in the TOMCAT transport model and associated errors in the wetland $CH_4$ emissions**
This comment was done by the two reviewers. The authors responded that several mitigation steps were taken including using model versions that have been previously used in other published works and that were consistently evaluated well against priors in $CH_4$ inversion exercises. Additionally, the authors suggest that the uncertainties generated in the TOMCAT atmospheric transport and chemistry model are likely larger than those introduced by not considering non-wetland emissions in JULES, and that quantitatively the uncertainties in emissions from e.g., biomass burning have shown to be much smaller than the uncertainties in wetland methane emissions, hence not strongly interfering in the modeled emissions.

The authors added a paragraph to discuss the impact of the assumptions taken, and performed an additional analysis in three regions of the world that are not typically dominated by wetlands. A comparison of detrended methane seasonal cycles between the satellite and model data showed a good agreement between both data sets, thus evidencing that non-wetland emissions can also be indirectly well constrained by the model.

I found that this approach to respond this point was well taken and well evidenced.

**2) On scaling annual emissions to the Global Methane Budget**

The authors scaled the annual emissions of 2000 to 180 Tg $CH_4$ $yr^{-1}$ to keep consistency with the reported value in Saunois et al., 2016 for wetland emissions. The authors argue that this step is necessary because the wetland area masked by SWAMPS in the JULES emissions is smaller giving as a result "unrealistically low methane emissions". The authors included a paragraph to clarify this step requested by referee #1.

However, in this added paragraph by the authors they mention: "the time series of annual wetland emissions of **each** ensemble member is separately scaled …". Wouldn't this scaling procedure needed to be done **only** for the ensemble members that were masked with SWAMPS? And not in **each** ensemble member. This is unclear and needs to be indicated more precisely or the added paragraph need to be moved to section 2.3.4

Also, it will be good to add a quantity to the expression "unrealistically low methane emissions" to give further validity to this approach in this particular set up.

**3) Comparison of GOSAT vs JULES**

Referee #2 points out the challenges in comparing GOSAT data to model surface emissions due to the potential of the satellite data to account for other methane sources that are not necessarily located in the region where the satellite is passing by, and also in relation to the problems with the seasonal coverage for specific regions. The authors provided an acceptable response referring to the technique utilized in GOSAT which uses only shortwave infrared and hence is limited only to land areas that emit such wavelength (i.e. wetlands).

Regarding the comparison of the two data sets shown in Figure 5, it is shown the difference in the amplitude between wetland seasonal cycles from the model emissions to that of the GOSAT data. From this figure, it is clear that GOSAT generally provides higher $CH_4$ emissions (about by 15 ppb) than the JULES simulated values in all the presented regions.
The color bar indicates the correlation coefficient between these two data sets, but: how is it possible to obtain a higher correlation coefficient in areas where the difference between the data sets is larger and with higher uncertainty (given by the whiskers in the bars)?
For example, the Pantanal region shows a correlation coefficient higher than 0.8 and has the highest range of difference and uncertainty of all the regions, and the South Tropics and S.E. Australia shows a close to zero difference but a correlation coefficient below 0.2. Is this a numerical artifact because the amplitude of the seasonal cycle is small or because the apparent time shift (Fig. 4) between the seasonal cycles of JULES and GOSAT? This should be make clearer in the text.

Also, it should be indicated why this time shift between the two data sets is happening.

**4) Regarding the atmospheric forcing data**
The authors concluded that the WFDEI data performed better than ERA-Interim in terms of wetland extent. The WFDEI data is based on ERA-Interim but its precipitation field is biased corrected using observational data, hence it is expected this improvement especially in the tropical areas.
However, WFDEI data is only available until December 2016 (and ERA-Interim until August 2019, and suppressed by ERA5), hence limiting the period of simulations using that forcing data.
According to NCAR, WFDEI is only available until December 2016 (https://rda.ucar.edu/datasets/ds314.2/), and the simulations using JULES in this manuscript were done until 2017, how the authors obtained forcing year for 2017?

Also, if model simulations using WFDEI are limited until 2016 (or 2017), this should be mentioned in the manuscript and, what recommendations can the authors provide for further studies that require improved wetland extent in simulations for years 2017 onwards? A test with ERA5 which has an improved precipitation field compared to ERA-Interim might be a way forward.

**Minor comments:**

The minor comments from both referees were answered accordingly and changes were implemented in the revised manuscript.

Own few minor comments:

Figure 1 – This is rather a table and not a figure, and in row 3 replace the expression "q10" by "$Q_{10}$"

Section 2.3.3 - Will be good to explicitly know why 3.7 and 5.0 were used as $Q_{10}$ values

---

## Author Response (AR2)

**Author Comments**

**Colour Key:** Reviewer Comment     Our Response     New Manuscript Text

**Reviewer 1**

We thank the reviewer for their comments and appreciate them taking the time to review our study. After this second round of reviews, we thank the reviewer for accepting the manuscript as is.

We would still like to address one issue as we believe that they were correct to raise it but it's something that we unfortunately cannot address at this stage.

Reviewer 1 and the Editor have both made the point, which we absolutely agree with, that the code should be in a more freely available location without requiring registration. Unfortunately, due to the current license of the JULES model, we are unable to put neither the full code nor key modules into any sort of openly available repository as the licensing terms prohibit this. Efforts are underway to improve this situation and GMD recently agreed that the JULES repository is an acceptable code source (see e.g. https://gmd.copernicus.org/preprints/gmd-2022-139/).

**Reviewer 2**

We thank the review for their comments, in taking part in this round of reviews and for recommending publication subject to the amendments below.

**RE: Emission scaling.** Wouldn't this scaling procedure needed to be done only for the ensemble members that were masked with SWAMPS? And not in each ensemble member. This is unclear and needs to be indicated more precisely or the added paragraph need to be moved to section 2.3.4

We agree that this is unclear and have updated the text as follows:

All JULES simulations, regardless of whether they have been further constrained with a wetland mask or not, are scaled such that the global total wetland methane annual emissions for 2000 has a value of 180 Tg/yr to ensure consistency with the best estimate of this value from Saunois et al. (2016).

Also, it will be good to add a quantity to the expression "unrealistically low methane emissions" to give further validity to this approach in this particular set up.

We have added clarification to the text on what we mean here and agree that this was poor phrasing on our part,

The scaling is most important when applied to the SWAMPS-based ensemble members as the geographic masking of the JULES wetland area with the SWAMPS data would otherwise result in reduced global emissions, below a level consistent with Saunois et al. (2016).

**RE: Figure 5** The color bar indicates the correlation coefficient between these two data sets, but: how is it possible to obtain a higher correlation coefficient in areas where the difference between the data sets is larger and with higher uncertainty (given by the whiskers in the bars)?

We will try to make this clearer in the text with the following explanation:

The bars indicate the difference in the **amplitude** of the seasonal cycle for each year. The colours indicate the correlation coefficient of the time series. It is entirely possible to have highly-correlated time series where the amplitude of the signal is different (e.g. two perfectly in-sync cycles but one with double the amplitude of the other). This is the case for e.g. Pantanal where the seasonality between JULES and GOSAT matches well, but the amplitude of the seasonal cycle is much larger in GOSAT than JULES. Conversely, there are regions where the maximum amplitude difference is small but the data is out of phase leading to a poor correlation, e.g. Indonesia.

Also, it should be indicated why this time shift between the two data sets is happening.

It's not so much a time shift as just a poor correlation. Section 5 is intended to explore these cases in detail, but we will add a sentence to the start of this section to make this clearer.

These regions were selected as they were found to exhibit particularly poor correlation coefficients between GOSAT and JULES, suggesting issues with the timing of the seasonal cycle.

According to NCAR, WFDEI is only available until December 2016 (https://rda.ucar.edu/datasets/ds314.2/), and the simulations using JULES in this manuscript were done until 2017, how the authors obtained forcing year for 2017?

WFDEI data has subsequently been extended beyond the dataset above (i.e. 2017 data was available).

What recommendations can the authors provide for further studies that require improved wetland extent in simulations for years 2017 onwards? A test with ERA5 which has an improved precipitation field compared to ERAInterim might be a way forward.

We have also added in the following statement

We would expect our conclusions regarding the strong performance of WFDEI meteorology to also apply to the updated WFDE5 data (based on ERA-5), detailed in Cucchi et al., 2020. Future will work assess simulations driven by these inputs.

Cucchi, M., Weedon, G. P., Amici, A., Bellouin, N., Lange, S., Müller Schmied, H., Hersbach, H., and Buontempo, C.: WFDE5: bias-adjusted ERA5 reanalysis data for impact studies, Earth Syst. Sci. Data, 12, 2097–2120, https://doi.org/10.5194/essd-12-2097-2020, 2020.

Figure 1 – in row 3 replace the expression "q10" by "Q10"

Thanks, noted and changed to $Q_{10}$.

Section 2.3.3 - Will be good to explicitly know why 3.7 and 5.0 were used as Q10 values

These values were chosen based on the work of Gedney et al. (2019) who tested values of 3, 3.7 and 4.7 as low/middle/upper estimates, themselves based on Turetsky et al. (2014).

We have clarified this in the text.

We chose $Q_{10}$ values of 3.7 and 5, based on the work of Gedney et al. (2019) who tested values of 3, 3.7 and 4.7 as low/middle/upper estimates, themselves based on Turetsky et al. (2014).